# Tracing nerve fibers with volume electron microscopy to quantitatively analyze brain connectivity

Marta Turegano-Lopez[1,2], Felix de las Pozas [ID][3], Andrea Santuy[4], Jose-Rodrigo Rodriguez[1,2,5],
Javier DeFelipe [ID][1,2,5] & Angel Merchan-Perez [ID][1,2,6] [✉]

The highly complex structure of the brain requires an approach that can unravel its connectivity. Using volume electron microscopy and a dedicated software we can trace and measure all nerve fibers present within different samples of brain tissue. With this software tool, individual dendrites and axons are traced, obtaining a simplified "skeleton" of each fiber, which is linked to its corresponding synaptic contacts. The result is an intricate meshwork of axons and dendrites interconnected by a cloud of synaptic junctions. To test this methodology, we apply it to the stratum radiatum of the hippocampus and layers 1 and 3 of the somatosensory cortex of the mouse. We find that nerve fibers are densely packed in the neuropil, reaching up to 9 kilometers per cubic mm. We obtain the number of synapses, the number and lengths of dendrites and axons, the linear densities of synapses established by dendrites and axons, and their location on dendritic spines and shafts. The quantitative data obtained through this method enable us to identify subtle traits and differences in the synaptic organization of the samples, which might have been overlooked in a qualitative analysis.

The brain is a highly complex structure, with numerous areas that differ in their cellular composition and connectivity patterns. Point-to-point connectivity is critical for the understanding of brain function[1], so defining the organizational principles of neuronal networks is currently a key—and challenging—goal in neuroscience[2,3]. At present, there are powerful methods to trace the connectome in macro- (long-distance connections) and meso-circuits (intrinsic or local connections). These methods include classical tracing methods[4], as well as molecular/genetic/physiological approaches and imaging techniques, such as two-photon imaging and optogenetic techniques[5] in addition to methods that enable integrated multiscale circuit mapping with excellent resolution and accuracy[6]. However, it is important to bear in mind that connectivity visualized at the light microscope level is generally rather limited (e.g., connections between brain regions) and, in most cases, point-to-point connections between local neurons and between neurons or afferent fibers cannot be accurately determined. This is because when a given labeled axon is seen in contact with another labeled neuronal element, this does not necessarily imply that there is a synaptic junction between them, since axonal boutons are adjacent to several possible synaptic targets of which only those that are labeled are visible. In addition, not all axonal boutons establish synaptic contacts, while some boutons may in fact establish several synapses[7]. Thus, the visualization of a labeled terminal in close apposition with a given neuronal element can only be considered a putative synaptic contact. Consequently, the available connectome diagrams lack precision, leading to a proposal that the term "connectome" be used to refer to the map of connections at the macroscopic and mesoscopic levels, with "synaptome" being used for the map at the ultrastructural level[1].

In the cerebral cortex, most synapses are found in the neuropil—the complex matrix of interwoven cytoplasmic processes of nerve and glial cells (dendrites, axons and glial processes) in which the cell bodies and blood vessels lie embedded[8]. Consequently, investigating the synaptic organization in the neuropil is crucial for understanding connectivity. However, this task is technically challenging, since it usually requires extensive 3D reconstructions of brain tissue at the electron microscopy level and advanced computational methods[9–17]. At present, full reconstruction of whole brains is only possible in some invertebrates or for relatively simple nervous

[1]Laboratorio Cajal de Circuitos Corticales, Centro de Tecnología Biomédica, Universidad Politécnica de Madrid, Pozuelo de Alarcón, 28223 Madrid, Spain. [2]Centro de Investigación Biomédica en Red de Enfermedades Neurodegenerativas (CIBERNED), Instituto de Salud Carlos III, 28031 Madrid, Spain. [3]Visualization & Graphics Lab (VG-Lab), Universidad Rey Juan Carlos, C/Tulipán S/N, Móstoles, 28933 Madrid, Spain. [4]Department of Basic Sciences, Universitat Internacional de Catalunya (UIC), San Cugat del Vallès, 08195 Barcelona, Spain. [5]Instituto Cajal, Consejo Superior de Investigaciones Científicas (CSIC), Avda. Doctor Arce, 37, 28002 Madrid, Spain. [6]Departamento de Arquitectura y Tecnología de Sistemas Informáticos, Universidad Politécnica de Madrid, Pozuelo de Alarcón, 28223 Madrid, Spain. A preprint has been uploaded at BioRxiv: https://www.biorxiv.org/content/10.1101/2023.09.26.559505v1. [✉]e-mail: angel.merchan@upm.es

systems. Indeed, even for a small mammal like the mouse, it is impossible to fully reconstruct the brain at the ultrastructural level since the magnification needed to visualize synapses yields relatively small images (in the order of tens of $\mu m^2$). To overcome this problem, different approaches have been proposed to study synaptic connectivity at the ultrastructural level (nano-connectivity) in the mammalian brain, with the gold standard being volume electron microscopy[18–20].

Some scientists consider that the best strategy is to study a relatively large volume of tissue to obtain saturated or dense reconstructions in a given region. Even if only one individual is examined using this approach due to the technical difficulties associated with this technology, it is considered the most advanced technology available for connectomic studies owing to the impressive view of the 3D ultrastructure of the tissue obtained (see, for example, references[14,21,22]). However, we believe that volume electron microscopy can be utilized more effectively by employing an alternative or complementary approach. This involves examining several smaller volumes of brain tissue from the region or regions of interest, and obtaining quantitative information from them that can be later analyzed and compared. Here we propose a method that does not require full reconstruction, but provides a simplified representation of the brain sample without losing information about its connectivity at the synaptic level. Briefly, we obtain stacks of serial images using focused ion beam milling and scanning electron microscopy (FIB-SEM)[10,11]. We used Espina software[23] to visualize, navigate and segment the stacks of sections. Instead of performing a full reconstruction of dendrites and axons, we have developed a tool in Espina to trace the course of axons and dendrites in 3D, obtaining a simplified skeleton of each nerve fiber and its corresponding synaptic contacts. The final result, once all fibers and connections have been traced, is a schematic representation of the fiber composition of the tissue and its connectivity. To explore the utility of this approach, we have formulated the following hypothesis: qualitative diversity in the cellular composition and connectivity of different brain regions should translate into quantitative differences in synaptic numbers and/or their distribution between dendrites and axons. To test this hypothesis, we have selected three regions of the adult mouse brain whose cellular composition and connectivity are very different: the stratum radiatum (SR) of the hippocampus (CA1 field), and layers 1 and 3 (L1 and L3) of the primary somatosensory cortex, in the hindlimb representation area (S1HL).

Once we have traced all the dendrites and axons, the software provides a set of quantitative parameters regarding the skeletons and their connections. These parameters include the number and proportions of excitatory (asymmetric) and inhibitory (symmetric) synapses; the numbers of dendrites and axons; the proportion of excitatory and inhibitory axons; the linear densities of synapses established on dendritic spines and dendritic shafts; the linear densities of synapses established by excitatory and inhibitory axons; and the lengths of the different types of dendrites and axons in a given tissue sample (Fig. 1 and Supplementary Movie 1). Taken together, these data provide a comprehensive view that can be used to characterize any given region of the brain and will help to understand brain connectivity at the synaptic level. In addition, the structural complexity of different regions can be analyzed and compared quantitatively.

## Results

In the following paragraphs, we present and compare quantitative data regarding synapses, dendrites and axons, as well as their connectivity in these three locations.

### Identification and segmentation of synapses

Synaptic junctions were identified and segmented in three stacks of serial sections obtained from the stratum radiatum of the hippocampus (SR), and layers 1 and 3 of the somatosensory cortex (L1 and L3, respectively). The number of sections per stack was 305 in SR and L1 and 263 in L3, which corresponds to a volume per stack of 549 $\mu m^3$ in SR and L1, and 473 $\mu m^3$ in L3. The study was performed in the neuropil, composed of axons, dendrites, and glial processes, so cell bodies or blood vessels were not present in the

imaged volume of tissue. Synaptic junctions were identified based on several morphological features[11,24]. Presynaptic elements show an accumulation of synaptic vesicles close to the presynaptic membrane, which shows an electron-dense thickening. The postsynaptic membrane also shows an electron-dense thickening, the postsynaptic density (PSD), which lies in front of the presynaptic thickening and is separated from it by a narrow synaptic cleft. If the synaptic junction is sectioned obliquely, the synaptic cleft may not be visible, but even when this occurs, the synaptic vesicles and membrane thickenings can be clearly identified in the series of images (Fig. 2a–d). The segmentation algorithm implemented in Espina software applies a threshold to the gray-level scale of the serial images, so the segmented object comprises the pre- and postsynaptic densities, which are the darkest elements of the synaptic junction (Fig. 2e–h). Since the segmentation process is performed in 3D, the final result is a cloud of segmented synaptic junctions inside the volume of brain tissue (Fig. 2i–l).

Synapses were classified into two groups according to the appearance of their PSD[25,26]. One of these types shows a prominent PSD, so they are termed asymmetric synapses (AS), due to the contrast between the relatively faint presynaptic thickening and the thick, electron-dense PSD. In the other type of synapses, the PSD is less marked, with a thickness that is similar to the presynaptic thickening; they are thus termed symmetric synapses (SS). In the cerebral cortex, AS are generally glutamatergic, excitatory synapses, while SS are mostly GABAergic, inhibitory synapses[27–29]. It should be emphasized that the classification of any given synapse as AS (excitatory) or SS (inhibitory) was not based on the visualization of single sections. Rather, it relied on the examination of all serial sections in which each synaptic junction was visible (Fig. 2a–h).

### Number of synapses

Once all of the synaptic junctions had been segmented, we counted the number of synapses inside an unbiased three-dimensional counting frame to obtain the density of synapses (number of synapses per cubic micron). For these calculations, we applied a correction factor to account for the tissue shrinkage that normally occurs during processing for electron microscopy (see Methods section). The total synaptic density in the SR (2.76 synapses/$\mu m^3$) was much higher than in L1 or L3 (1.80 and 1.08 synapses/$\mu m^3$, respectively) (Fig. 2m, Supplementary Table S1). Most synapses were AS (up to 97.10% in the SR). The highest SS density was found in L1 (0.20 synapses/$\mu m^3$; 11.11% of all synapses). It is interesting to note that the absolute densities of SS in the SR and L3 were similar (0.08 and 0.07 synapses/$\mu m^3$, respectively), but the corresponding percentages were very different (2.90% and 6.09% of all synapses, respectively) (Fig. 2n, Supplementary Table S1). The differences in the proportions of AS and SS were statistically significant between the SR and L1 (Chi-square; $p < 0.05$).

### Dendrites with spines and smooth dendrites

Once all synaptic junctions had been segmented, we proceeded to identify and trace dendrites and axons. The paths followed by individual nerve fibers were interactively traced in 3D, making use of the navigation capabilities of Espina software. A trace was drawn for every single fiber, representing its trajectory within the volume of tissue. Any fiber segment postsynaptic to at least one synaptic junction was tagged as a dendrite. Dendrites with spines were the most common and were tagged as "spiny" dendrites; in these cases, we traced the dendritic shaft and the spines associated with it (Fig. 3a,b; Supplementary Movie 1). Dendrites without spines were tagged as "smooth" dendrites. Every dendrite was linked to its corresponding synapses. Thus, the final result of the complete tracing of any given dendrite comprised of the dendritic shaft, its spines, and its associated synapses (Fig. 3c; Supplementary Movie 1).

Spiny dendrites were clearly more numerous than smooth dendrites in the three locations. In SR, 91.04% of all dendrites were spiny dendrites. In L1 and L3, respectively, 93.10% and 93.75% of all dendrites were spiny dendrites. (Fig. 3d). Therefore, under 9% of all dendrites in the three locations studied were dendrites without spines, or smooth dendrites. The differences in the proportions of spiny and smooth dendrites between the three

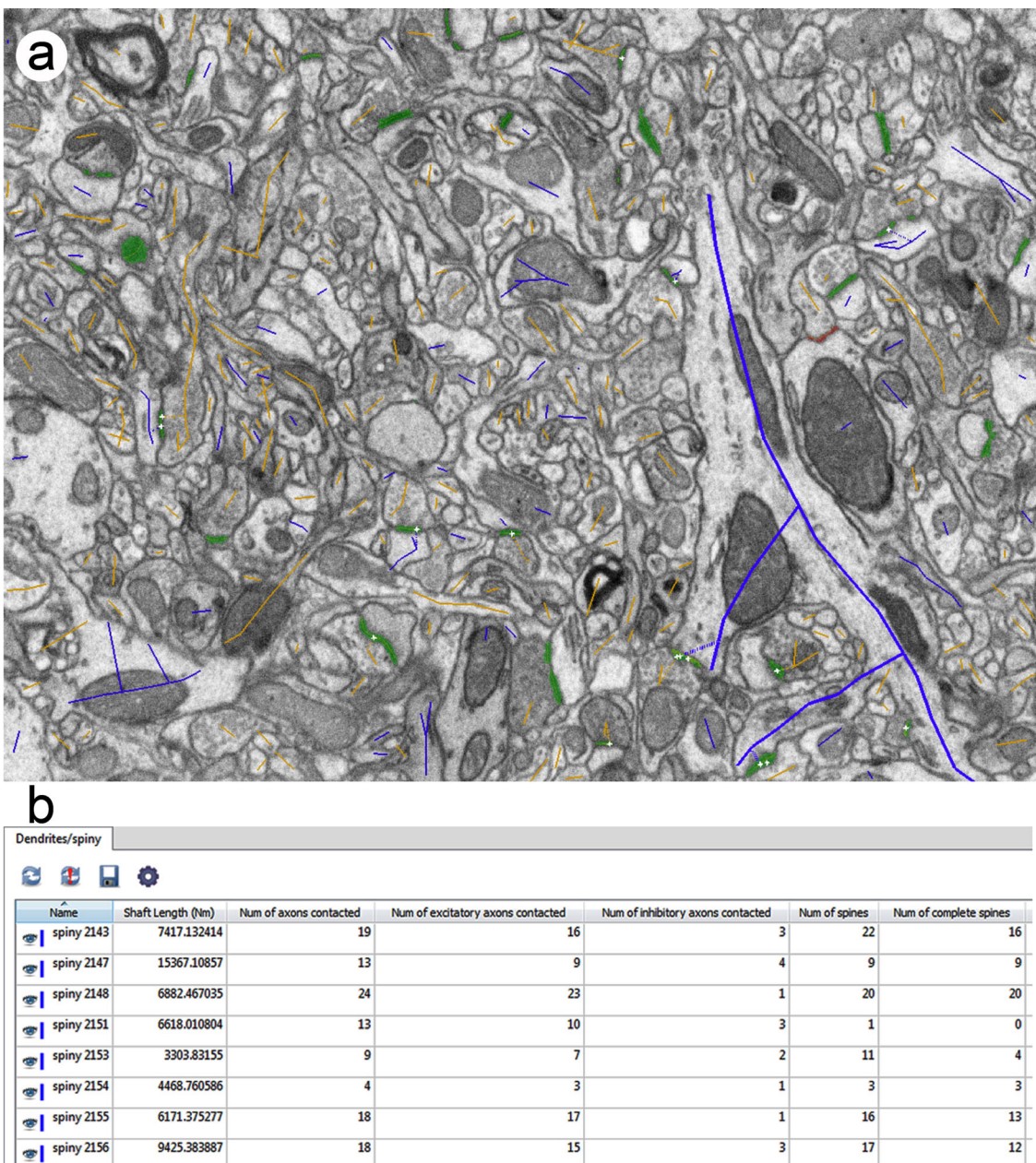

**Fig. 1 | Tracing dendrites and axons. a** Snapshot of an image from a series obtained with FIB-SEM. Dendrites and axons have been traced with Espina software, in blue and orange, respectively. Tracing is performed in 3D, using the whole stack of serial images, although only one image is shown here. See Supplementary Movie 1 to see how tracing is performed by navigating the stack in 3D. Synaptic junctions have been connected to their parent dendrites and axons. Synaptic junctions have been segmented in green. **b** The software provides information from each individual trace. In this example, quantitative information includes the length of several dendrites, number of axons contacted and number of spines.

locations were not significant at $p = 0.05$ (Chi-square). Synapses were unequally distributed between spines and dendritic shafts. Most synapses were established on dendritic spines in all three locations (84.65%, 74.81% and 80.45% in SR, L1 and L3, respectively), followed by synapses on the shafts of spiny dendrites and smooth dendrites (Fig. 3e; Supplementary Table S2). Note that the proportion of synapses on smooth dendrites was clearly higher in the SR than in L1 and L3. On the contrary, the proportion of synapses on the shafts of spiny dendrites was much lower in the SR than in L1 or L3. The differences in the distributions of synapses between the SR and the other two regions were statistically significant (Chi-square; $p < 0.001$).

The linear density of synapses was calculated as the number of synapses per micron of dendritic shaft (Fig. 3f; Supplementary Table S3). The mean linear density of synapses on spiny dendrites was higher in the SR

(3.33 synapses/μm) than in L1 and L3 (1.77 and 1.88 synapses/μm, respectively) and the differences were statistically significant (KW, Bonferroni correction, $p < 0.001$). The linear density of synapses on smooth dendrites was also the highest in the SR (1.84 synapses/μm) when compared with L1 and L3 (0.65 and 0.66 synapses/μm, respectively), but in this case, the number of smooth dendrites was small and the differences were not statistically significant (KW $p > 0.1$). Within each region, the linear densities of synapses on spiny dendrites were always much higher than on smooth dendrites, and the differences were statistically significant in the SR and L1 (MW, $p < 0.05$ and $p < 0.01$, respectively) (Supplementary Table S3).

For spiny dendrites, we also calculated the linear density of synapses on spines and shafts independently (Fig. 3f; Supplementary Table S3). The mean linear density of synapses on spines was not homogenous between the

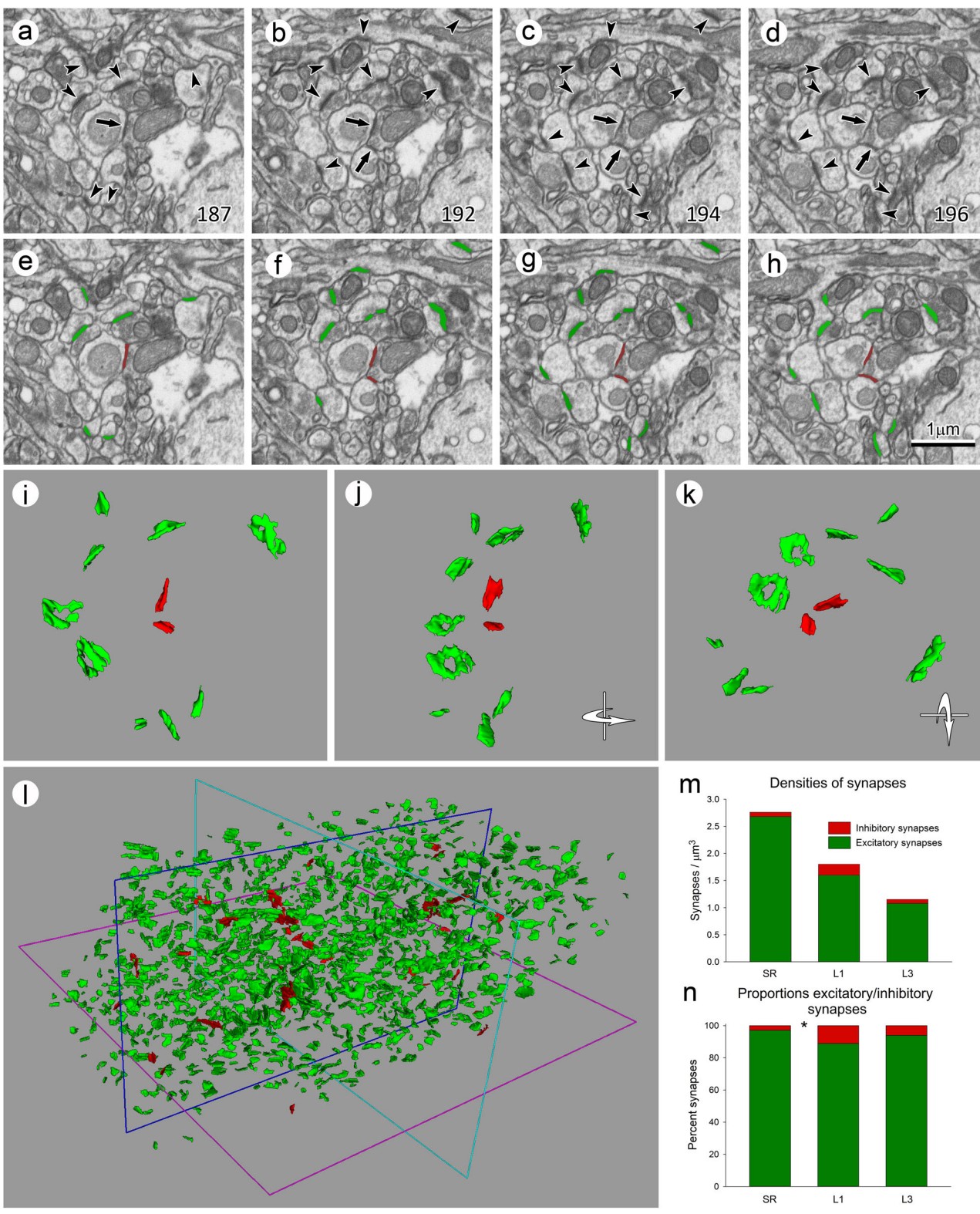

three regions analyzed (KW, $p < 0.001$). It was much higher for dendrites in the SR (3.08 synapses/μm) than for dendrites in L1 or L3 (1.41 and 1.56 synapses/μm, respectively; Supplementary Table S3). Differences between the SR and L1 or L3 were statistically significant (KW, Bonferroni correction, $p < 0.001$), but differences between L1 and L3 were not ($p > 0.1$).

The mean linear density of synapses established on the shafts of spiny dendrites ranged from 0.25 synapses/μm in SR to 0.36 synapses/μm in L1,

although the differences were not statistically significant (KW, $p = 0.326$). The mean linear density of synapses on smooth dendrites was higher in the SR (1.84 synapses/μm) than in the other two locations (0.65 and 0.66 synapses/μm in L1 and L3, respectively; Supplementary Table S3), but the differences were not significant (KW, $p = 0.209$). Within each region, the linear densities of synapses on dendritic spines were always the highest, followed by smooth dendrites, which in turn were higher than the linear

**Fig. 2 | Segmentation of synapses in a stack of serial sections. a–d** Four non-consecutive serial images were acquired from the stratum radiatum (SR) using focused ion beam milling and scanning electron microscopy (FIB-SEM). The number in the bottom right corner indicates the position of each microphotograph in the series. Excitatory synapses (arrowheads) are identified by the presence of a thick postsynaptic density (PSD). Inhibitory synapses (arrows) are also visible, showing a thin PSD. **e–h** The same micrographs once the synapses have been segmented in 3D with Espina software. Excitatory synapses have been represented in green and inhibitory synapses in red. **i** Three-dimensional rendering of the group of synaptic junctions previously shown in (**a**) to (**h**). The reconstructed synaptic junctions have been rotated through a vertical and a horizontal axis in (**j**) and (**k**), respectively. **l** Three-dimensional rendering of all synapses that are present in the whole stack of serial sections; excitatory synapses (green) predominate over inhibitory synapses (red). **m** The number of synapses per cubic micron measured in stacks of serial sections from the SR, and from layers 1 and 3 of the somatosensory cortex (L1 and L3). The global density of synapses was highest in the SR and lowest in L3. The highest density of inhibitory synapses was found in L1. **n** Proportions of excitatory and inhibitory synapses in the three brain regions. Excitatory synapses clearly outnumber inhibitory synapses in all regions; L1 shows the highest proportion of inhibitory synapses, and SR the lowest. The asterisk indicates that the differences in the proportions of AS and SS between SR and L1 were statistically significant (Chi-square; $p < 0.05$). See also Supplementary Table S1.

densities of synapses on the shafts of spiny neurons; the differences between the linear densities of synapses on spines and shafts were statistically significant in the three regions studied (KW, Bonferroni correction, $p < 0.001$) (Fig. 3f; Supplementary Table S3). It is interesting to note that we found no correlation between the density of synapses on spines and the density of synapses on the shaft of individual dendrites ($R^2 = 0.1475$, 0.0003 and 0.0036 in the SR, L1 and L3, respectively).

Linear densities of synapses were not distributed homogeneously between individual dendrites. In spiny dendrites, it was possible to find dendritic segments with very different linear densities of synapses in the three areas analyzed (Fig. 4a). This heterogeneity was the highest in the SR, where linear densities ranged between 0.20 and 6.49 synapses/µm (Fig. 4b). In L1 and L3, the frequency distributions were narrower, with ranges between 0.31 and 4.34 synapses/µm in L1, and 0.14 and 4.70 synapses/µm in L3 (Fig. 4b). Smooth dendrites also showed heterogeneous linear densities of synapses (Fig. 5).

### Excitatory and inhibitory axons

Any fiber that was presynaptic to at least one synapse was tagged as an axon. Axonal segments were further subclassified into excitatory, if they established AS; inhibitory, if they established SS; or myelinated axons, when they were covered by a myelin sheet. Each axon was represented by its corresponding tracing and its associated synapses (Fig. 6a). Excitatory axons clearly outnumbered inhibitory axons in all regions studied. In the SR, the proportion of excitatory axons was the highest, reaching 97.56% of all axons. In L1 and L3 91.59% and 94.21% of all axons were excitatory. The differences in the proportions of excitatory and inhibitory axons where statistically significant when the SR was compared with either of the other two brain regions (Chi-square; $p < 0.05$) (Fig. 6b).

Excitatory and inhibitory axons showed different preferences to establish synapses on spines or dendritic shafts. The most frequent type of synapses in all three locations studied were AS on dendritic spines (79.06% to 90.10%), followed by much lower percentages of AS on dendritic shafts (5.99% to 10.52%), SS on shafts (1.73% to 8.02%) and SS on spines (0.73% to 2.46%) (Fig. 6c; Supplementary Table S4). The differences between the three regions were statistically significant (Chi-square; $p < 0.001$).

This distribution implies a clear preference of excitatory axons for dendritic spines, since 88.26% to 93.62% of all AS were axospinous (Supplementary Table S5). On the contrary, inhibitory axons prefer dendritic shafts, although this preference is not so strong since 60–77% of SS were established on shafts while 23% to 40% of all SS were axo-spinous (Supplementary Table S5). Consequently, the vast majority of synapses on spines were AS (more than 97% of synapses on spines were AS). AS also outnumbered SS on dendritic shafts, although the proportions were different in the three locations studied (56.74% to 81.16% of synapses on dendritic shafts were AS; Supplementary Table S5). It is interesting to note that all SS that were established on spines were accompanied by another AS on the same spine, and 84.38% were established on the head of the spine, while the remaining 15.63% were established on the spine neck.

We calculated the linear density of synapses established by inhibitory and excitatory axons (Supplementary Table S6). For excitatory axons, the mean linear density of synapses in the SR (0.38 synapses/µm) was much higher than in L1 and L3 (0.23 and 0.24 synapses/µm, respectively), and this difference was statistically significant (KW, Bonferroni correction; $p < 0.001$) (Fig. 6d; Supplementary Table S6). For inhibitory axons, the mean linear densities were similar in the three locations (0.33; 0.30, and 0.26 synapses/µm, in SR, L1 and L3, respectively; KW, $p = 0.312$) (Supplementary Table S6). From a different point of view, when we compared the linear densities of excitatory and inhibitory axons within each region, we found that the differences were only statistically significant in L1 (MW, $p < 0.01$) (Fig. 6d; Supplementary Table S6).

As in the case of dendrites, linear densities of synapses were not homogeneous across different axons (Fig. 7a). Frequency histograms showed that linear densities of synapses of excitatory axons followed an asymmetric distribution, with a tail to the right (Fig. 7b). The widest range was found in the SR, where the linear densities of synapses of excitatory axons could be as low as 0.10 and as high as 1.31 synapses/µm. Narrower ranges were found in L1 (0.06–0.78 synapses/µm) and L3 (0.08–0.70 synapses/µm). Inhibitory axons also showed heterogeneous linear densities of synapses (Fig. 7a).

We addressed the question of the number of axons that established two or more synapses with the same dendrite. To make data comparable, we counted the number of axons that established at least two synapses, and the proportion of these that established two or more synapses with the same dendrite. For excitatory axons, we found that these proportions were 10.27%, 8.06% and 10.08% in the SR, L1 and L3, respectively, with no statistically significant differences between regions (Chi-square, $p = 0.711$). Most of these axons established multiple contacts on spines (97.50% of synapses were established on spines, and the remaining 2.50% were established on the shaft and one spine). One single excitatory axon in the SR established two synapses on the same smooth dendrite. Regarding inhibitory axons, the proportions of them with multiple synapses on the same dendrite showed a large variability (from 14.81% to 33.33%), probably due to the small number of inhibitory axons in our samples (Supplementary Table S7).

We also calculated the proportions of dendrites that established multiple synaptic contacts with the same axon. In the SR, 50.79% of dendrites established repeated contacts with one or more axons, while the percentages in L1 and L3 were 23.17% and 26.83%, respectively. Differences were statistically significant between the SR and L1 (Chi square, $p < 0.05$). In the SR, we found eight dendrites establishing multiple synapses with two or more (up to six) different axons. Only two of these dendrites were found in L1 and another two in L3 (Supplementary Table S7). The axons with multiple contacts were mostly excitatory, but we also found dendrites receiving multiple contacts from excitatory and inhibitory axons.

Most synapses were established *en passant*, while synapses established by terminal boutons were much less frequent. For excitatory axons, the proportions of *en passant* synapses were 95.44%, 73.26%, and 80.49% in the SR, L1, and L3, respectively, with the differences in the proportions being statistically significant between the three brain regions studied (Chi-square, $p < 0.005$). For inhibitory axons, 97.30%, 76.00% and 85.71% of the synapses were *en passant* in the SR, L1 and L3, respectively; in this case, the only statistically significant difference was found between the SR and L1 (Chi-square, $p < 0.005$). The differences in proportions of *en passant* and terminal

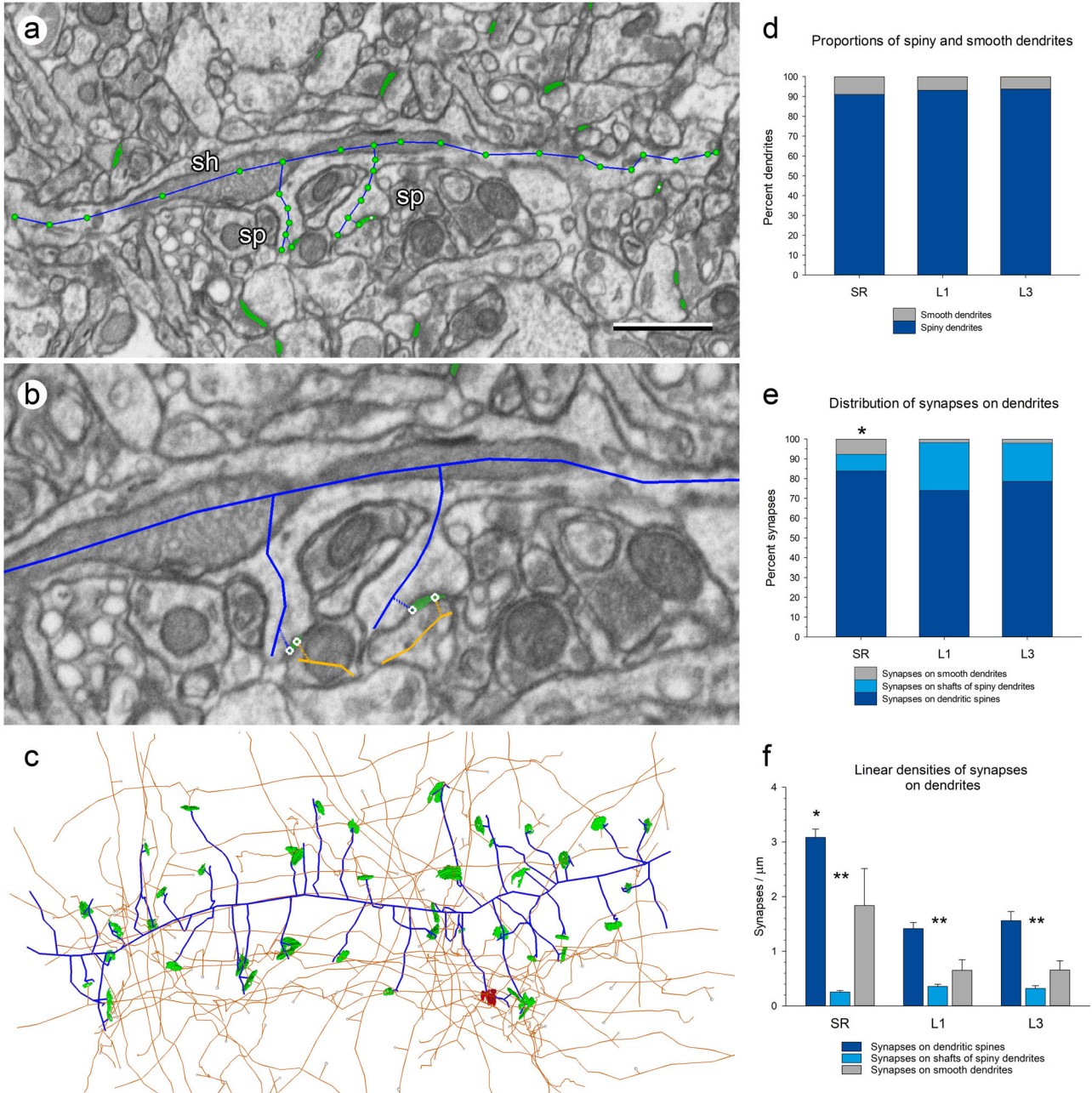

**Fig. 3 | Tracing dendritic segments. a** Panoramic view of a dendritic segment that has been traced with Espina software (blue trace). In this example, the dendritic shaft (sh) and two spines (sp) have been traced. Green profiles are excitatory synaptic junctions that have been previously segmented. **b** Higher magnification of the region of the dendrite where the two spines arise. The traces of each spine are connected to an excitatory synapse (green), which is in turn connected to its parent axon (orange). **c** Three-dimensional representation of the same dendrite (blue) once it has been completely traced across the whole series of FIB-SEM images. Excitatory (green) and inhibitory synapses (red) established on this dendritic segment, and their parent axons (orange) have also been represented. Scale bar: 1 μm in (**a**); 0.55 μm in (**b**); 1.25 μm in (**c**). **d** Proportions of dendrites with spines (spiny dendrites, dark blue) and dendrites without spines (smooth dendrites, gray) in the stratum radiatum of the hippocampus (SR), and layers 1 and 3 of the somatosensory cortex (L1 and L3). Smooth dendrites are always present, but they are scarce in the three regions. **e** Distribution of synapses on spines (dark blue), on the shafts of spiny dendrites

(light blue) and on smooth dendrites (gray). See also Supplementary Table S2. Asterisk: the differences in the distributions of synapses between the SR and the other two regions were statistically significant (Chi-square; $p < 0.001$). **f** Linear densities of synapses established on dendrites in the three regions, expressed as the mean number of synapses per micron of dendritic shaft (plus standard error of the mean). For spiny dendrites, the linear densities of synapses have been broken down into those established on spines (dark blue) and those established on the shafts (light blue). The average linear densities of synapses on smooth dendrites (gray) are in all cases between the densities of synapses on the shafts of spiny dendrites and the densities of synapses on spines. See also Supplementary Table S3. Asterisk: linear density of synapses on dendritic spines in the SR was higher than in L1 or L3 (KW, $p < 0.001$). Double asterisks: differences within each brain region between the linear densities of synapses on spines and shafts were statistically significant (KW, $p < 0.001$).

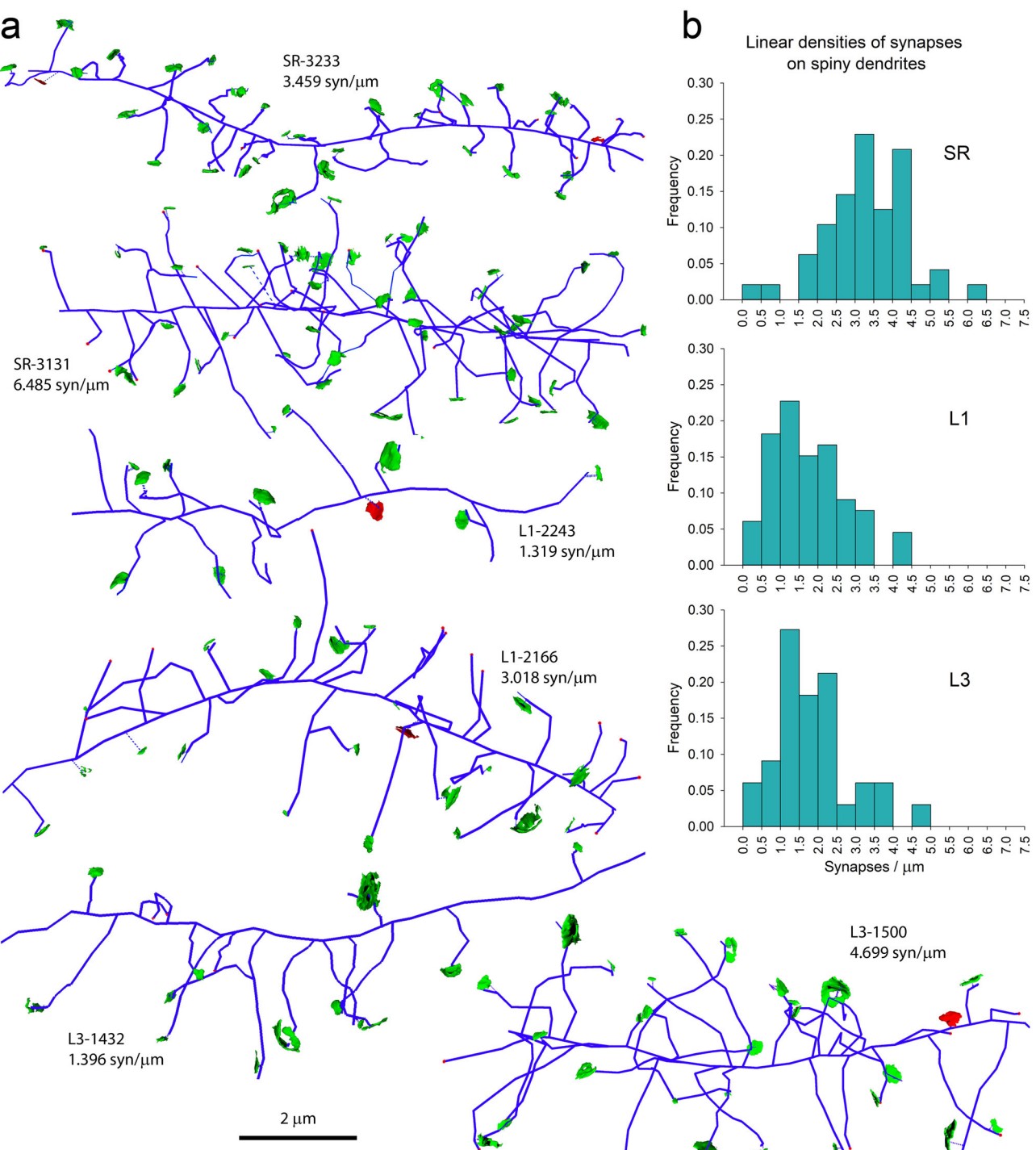

**Fig. 4 | Examples of spiny dendrites with different linear densities of synapses.** **a** Tracings of spiny dendrites (blue) from the stratum radiatum of the hippocampus (SR), and layers 1 and 3 of the somatosensory cortex (L1 and L3). Each tracing is identified by the region (SR, L1 or L3) followed by the identification number of the dendrite, and the linear density of synapses expressed in synapses per micron of dendritic shaft. Excitatory (green) and inhibitory (red) synapses that are established on each dendrite have also been represented. **b** Histograms of the distributions of the linear densities of synapses established on spiny dendrites of the three regions.

synapses between excitatory and inhibitory axons were not statistically significant within any of the three regions (Chi-square, $p > 0.1$) (Supplementary Table S8).

### Non-synaptic fibers
We also traced and measured the fibers that traverse the volumes without establishing any synapses. Typically, these are thin, unbranched fibers resembling axonal segments. However, due to the absence of synapses, their classification as dendrites or axons cannot be established with certainty. Therefore, we categorized them separately as "non-synaptic" fibers. In the SR and L1, these fibers exhibited multidirectional trajectories and accounted for 4.67% and 14.79% of the total length of nerve fibers, respectively. In L3, they were more numerous (31.49% of the total length of fibers) and displayed a preference for vertical orientation (Table 1, Fig. 8).

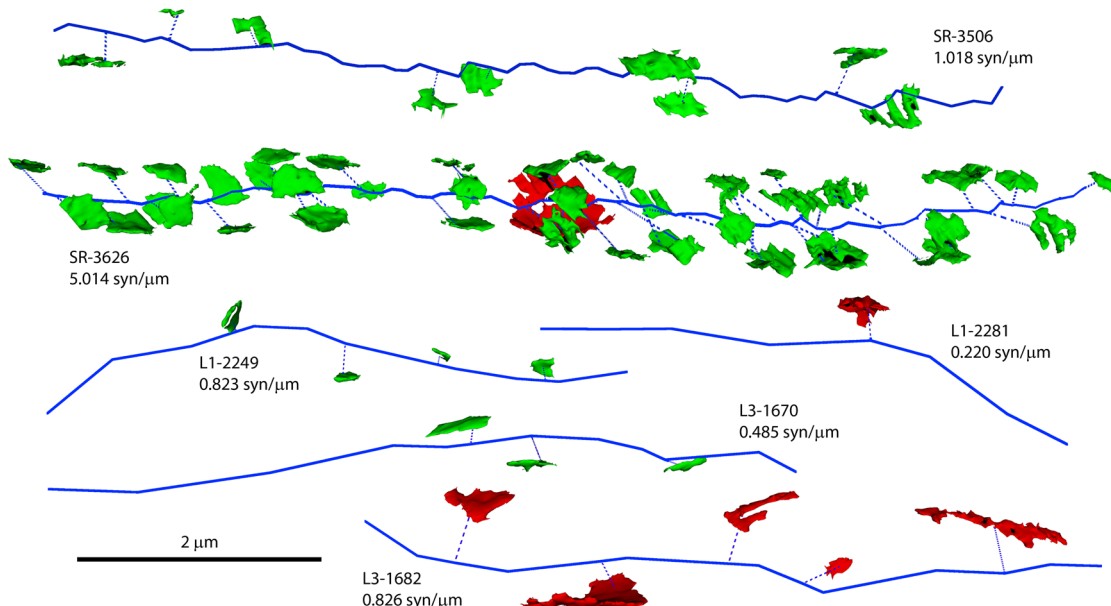

**Fig. 5 | Examples of tracings of smooth dendrites.** Several examples of dendrites without dendritic spines (blue) from the stratum radiatum of the hippocampus (SR), and layers 1 and 3 of the somatosensory cortex (L1 and L3). Excitatory (green) and inhibitory (red) synapses have also been represented. Each tracing is identified by the region (SR, L1 or L3) followed by the identification number of the dendrite, and the linear density of synapses expressed in synapses per micron of dendritic shaft.

## Overview

The final result of the complete segmentation and tracing of a stack of serial sections was a tangle of multiple axonal and dendritic segments interconnected by a cloud of synapses (Fig. 9a), providing a nanomap of synaptic connections. Despite the great complexity of the multiple fiber components and their connections, the different populations of fibers and synapses can be represented and studied independently, and any individual trace and its associated synapses can be singled out. For example, only inhibitory axons can be selected, or any given dendrite together with all axons that establish synapses with it—or a single axon with the dendrites that establish synapses with it. In this way, we were able to determine the relative proportions of the different types of dendrites and axons, and calculate the total lengths of dendrites and axons within a given sample of tissue.

The estimated total length of nerve fibers per cubic mm of neuropil ranged from 7546 m/mm$^3$ in L3, to 9378 m/mm$^3$ in L1 (Table 1; Fig. 9b; Supplementary Data 1). Non-synaptic fibers were the most variable category, ranging from 4.67% to 31.49% of the total length of fibers. If we only consider the fibers that establish synapses and were unambiguously identified as dendrites and axons, the estimated total length was similar in the SR and L1 (7793.73 and 7991.50 m/mm$^3$, respectively), while the total length of these fibers was lower in L3 (5170.00 m/mm$^3$) (Table 1; Fig. 9b).

Although there were differences between the three regions studied, there was a common pattern in the relative length of the different types of fibers, leaving aside non-synaptic fibers. Excitatory axons were, by far, the most common type in all locations, comprising between 56.15% and 82.70% of the total length of all nerve fibers (Table 1). These were followed by spiny dendrites (which accounted for 7.97% to 10.25% of the total length of fibers) and inhibitory axons (2.39–6.17% of all fibers). Finally, smooth dendrites and myelinated axons were scarce in our samples, with percentages around or below 1% of the total length of nerve fibers (Table 1; Fig. 9b; Supplementary Data 1).

## Discussion

We have compared the synaptic organization of the neuropil in three regions of the mouse brain that exhibit distinct cellular composition and connectivity: the stratum radiatum (SR) of the hippocampus (CA1 field), and layers 1 and 3 (L1 and L3) of the primary somatosensory cortex in the

hindlimb representation area (S1HL). The SR is mainly composed of the ramifications of the apical dendrites of pyramidal cells and Schaffer collateral/commissural axons[30,31]. It is almost devoid of neuronal somata, which correspond to GABAergic inhibitory interneurons[32,33]. L1 of the neocortex is mainly composed of the distal ramifications of dendrites of neurons located in deeper layers and axons that originate in deeper cortical layers, other cortical areas, and subcortical nuclei[34]. Neuronal somata are scarce in L1, with a small representation of excitatory Cajal-Retzius cells[35,36], while inhibitory GABAergic neurons are the most abundant neuronal type[37–40]. L3 is densely packed with small pyramidal neurons. It is composed of the basal an apical dendritic trees of the local neurons, the apical dendrites of cells located in deeper layers, and axons of heterogeneous origins[41]. In addition to the numerous excitatory pyramidal cells, lower numbers of inhibitory neurons can be found in L3[27,42,43].

Different volume EM methods allow us to obtain long series of images from consecutive sections of brain tissue, where subcellular structures, including synaptic junctions, can be visualized. However, the identification and annotation of synapses in different laboratories vary. Some studies provide assumption-based estimations. For example, synapses in the neuropil are sometimes classified as excitatory or inhibitory on the basis of whether they synapse onto dendritic spines or shafts, respectively[21,22,44,45], and they are annotated manually or using machine learning tools to train automated synapse classifiers to identify the pre- and postsynaptic component of each synapse (e.g., ref. 46). However, in our present study, the identification of synapses is based on the classical definition of synaptic contacts visualized at high resolution in serial sections. Strictly, a structure is identified as a synapse when the following elements are clearly recognized[8]: synaptic vesicles in the presynaptic axon terminal adjacent to the presynaptic density; a synaptic cleft (with electron-dense material in the cleft); and densities on the cytoplasmic faces in the pre- and postsynaptic membranes. There is a general consensus for classifying cortical synapses into asymmetric (or type I) and symmetric (or type II) synapses[25,26]. The main characteristic distinguishing these synapses is either a prominent or a thin postsynaptic density, respectively. In the cerebral cortex, excitatory glutamatergic synapses are of the asymmetric type, while inhibitory GABAergic synapses are symmetric[27–29,47]. These synapses can be segmented using semiautomatic or machine learning algorithms using specifically designed software such as Espina[23]. In this way, we can obtain—from any given stack

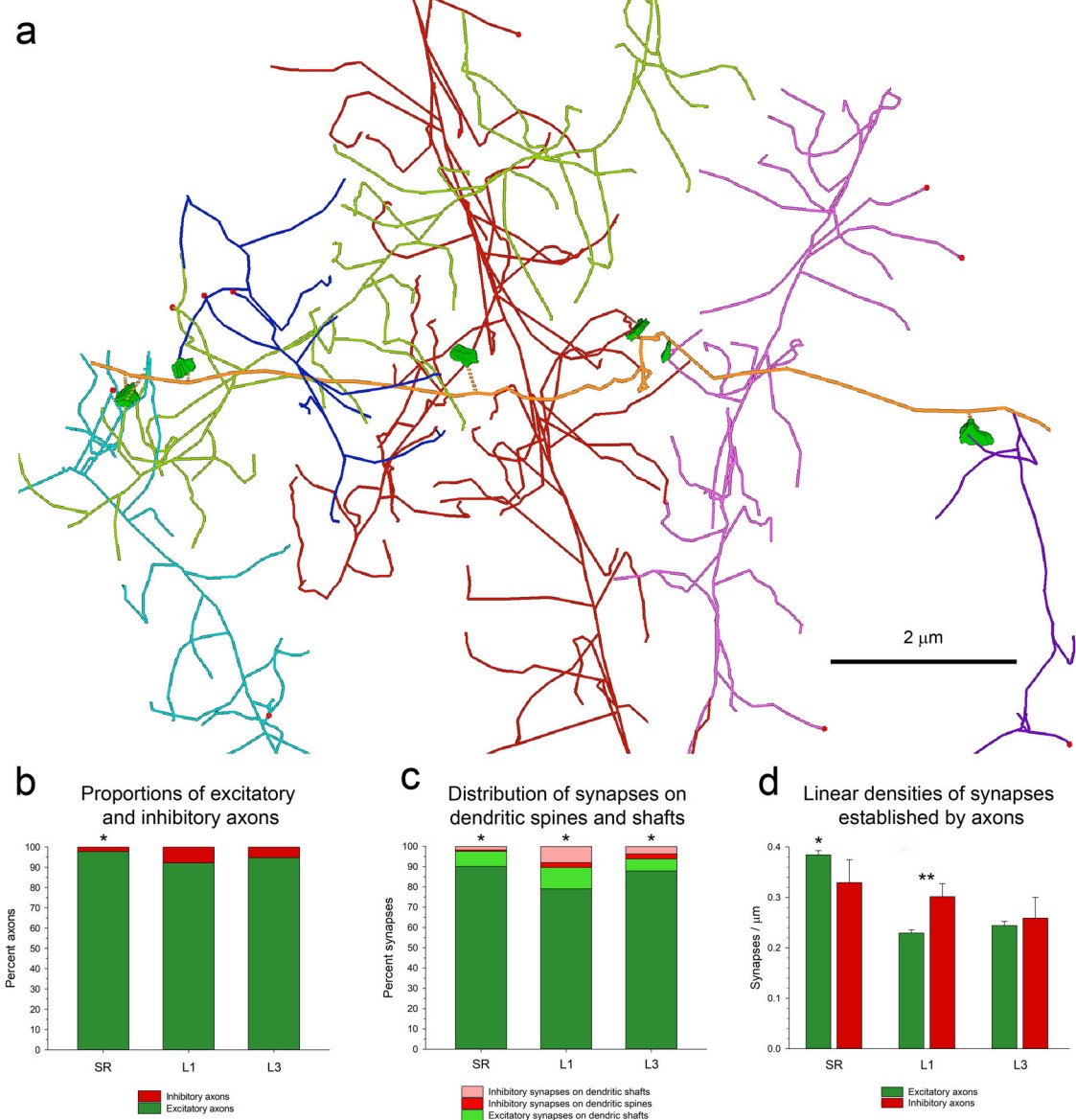

**Fig. 6 | Tracing axonal segments. a** An example of an excitatory axon (orange) that has been traced in the stratum radiatum of the hippocampus (SR). This axonal segment establishes excitatory synapses (green) with six different dendrites (represented in different colors). **b** Proportions of excitatory (green) and inhibitory (red) axons in the SR, and layers 1 and 3 of the somatosensory cortex (L1 and L3, respectively). Asterisk: the differences between the SR and the other two locations were statistically significant (Chi-square; $p < 0.05$). **c** Distribution of synapses between dendritic spines and shafts. Excitatory synapses in the SR, L1 and L3 have been broken down into those established on spines (dark green) and those established on dendritic shafts (light green). Inhibitory synapses on spines (red) and on dendritic shafts (pink) have also been represented. See also Supplementary Tables S4, S5. Asterisks: Differences between the three brain regions were statistically significant (Chi-square; $p < 0.001$). **d** Linear densities of synapses established by excitatory (green) and inhibitory (red) axons in the SR, L1 and L3. See also Supplementary Table S6. Asterisk: the linear density of synapses established by excitatory axons in the SR is larger than in either of the other two regions (KW, Bonferroni correction; $p < 0.001$). Double asterisk: differences between excitatory and inhibitory axons are statistically significant in L1 (MW, $p < 0.001$).

of images of the brain—information about the actual number of synapses; the proportion of excitatory and inhibitory synapses; morphological properties; spatial distribution; and preferred postsynaptic targets (dendritic spines and shafts). This information can be obtained from different brain regions, even from human brain (see below), and it is critical to identify principles of organization of brain connectivity at the local level, and to compare synaptic connectivity in normal and pathological conditions. In addition, these quantitative data are essential to generate realistic brain models.

Another important advantage of our study is the image resolution. For example, in the study of Motta et al., using SBEM[22], the voxel size of the 3D EM dataset was $11.24 \times 11.24 \times 28$ nm$^3$, whereas in our studies using FIB-

SEM, the image resolution is substantially higher—$5 \times 5 \times 20$ nm$^3$, that is, 5 nm/pixel in the xy plane and 20 nm in the z axis (equivalent to section thickness). This methodological difference holds critical significance from a connectomics perspective, as it enables the classification of excitatory and inhibitory synapses based on precise ultrastructural features. Our analysis avoids assumptions concerning synapse type definitions and postsynaptic elements. As a result, synaptic junctions can be unambiguously classified as excitatory or inhibitory, and their sizes and shapes can be accurately measured. This approach allows us to identify each individual synapse in our samples, enabling a direct determination of true synaptic density and types per unit volume, without relying on estimations through stereological methods or theoretical assumptions.

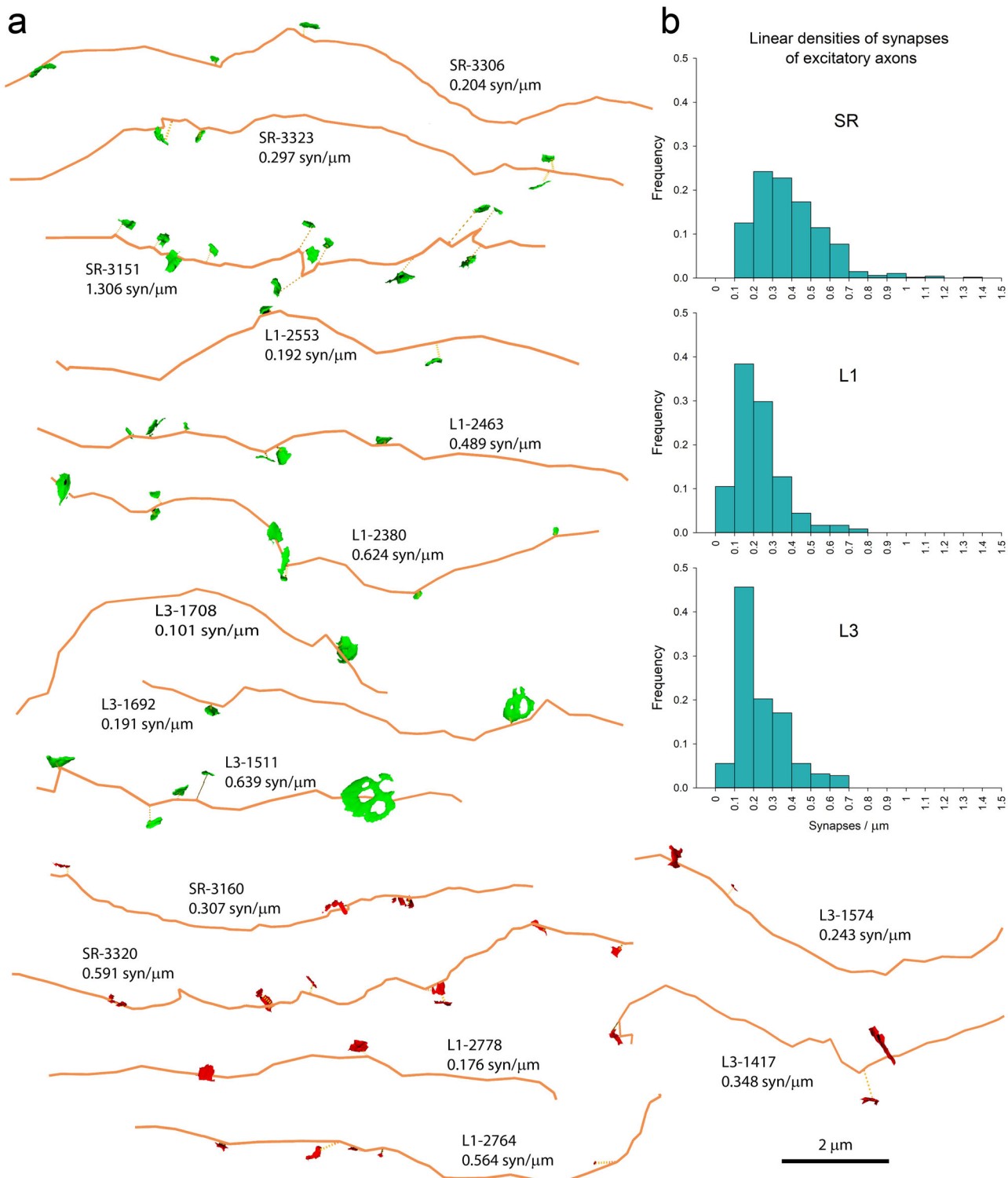

**Fig. 7 | Examples of excitatory and inhibitory axons. a** Tracings of axons (orange) from the stratum radiatum of the hippocampus (SR), and layers 1 and 3 of the somatosensory cortex (L1 and L3). Axons establish either excitatory synapses (green) or inhibitory synapses (red). Each axonal tracing is identified by the region (SR, L1 or L3), an identification number, and the linear density of synapses expressed as the number of synapses per micron. **b** Histograms of the distribution of the linear densities of synapses established by excitatory axons for each of the three regions.

The density of synapses that we have found in the SR (2.76 synapses/$\mu m^3$) is in line with a previous study performed with similar methodology in the mouse hippocampus (range: 2.06–2.88 synapses/$\mu m^3$, mean: 2.36 synapses/$\mu m^3$)[48]. The proportion of AS to SS in the present study (97.1/2.9) is also similar to the previously calculated proportion (97.5/2.5)[48], or the reported 3% of inhibitory synapses in thin dendrites, which predominate in

our sample[49]. The density of synapses in the rat SR is about 2.2–2.5 synapses/$\mu m^3$ [50–52]. In L1 and L3, we found synaptic densities of 1.80 and 1.08 synapses/$\mu m^3$, respectively, that is, clearly lower than in the SR. These data are in line with the synaptic densities described in the somatosensory cortex of the Etruscan shrew[53]. In the somatosensory cortex of the juvenile rat, the synaptic densities are lower, but the proportion of excitatory and

**Table 1 | Lengths of nerve fibers**

| Region | Length of non-synaptic fibers (m/mm³) | Length of excitatory axons (m/mm³) | Length of inhibitory axons (m/mm³) | Length of myelinated axons (m/mm³) | Length of spiny dendrites (m/mm³) | Length of smooth dendrites (m/mm³) | Total length of nerve fibers (m/mm³) |
|---|---|---|---|---|---|---|---|
| SR | 382.06 4.67% | 6761.19 82.70% | 195.14 2.39% | 17.72 0.22% | 731.79 8.95% | 87.89 1.08% | 8175.79 100% |
| L1 | 1387.28 14.79% | 6371.80 67.94% | 578.53 6.17% | 42.68 0.46% | 961.27 10.25% | 37.22 0.40% | 9378.79 100% |
| L3 | 2376.22 31.49% | 4237.03 56.15% | 282.96 3.75% | 21.16 0.28% | 601.56 7.97% | 27.30 0.36% | 7546.22 100% |

Lengths of the different types of neuronal processes in the stratum radiatum (SR), and layers 1 and 3 of the somatosensory cortex (L1 and L3). Data are given as the length of fibers in meters per cubic mm of tissue, and the percentage of the total length of nerve fibers in each region.

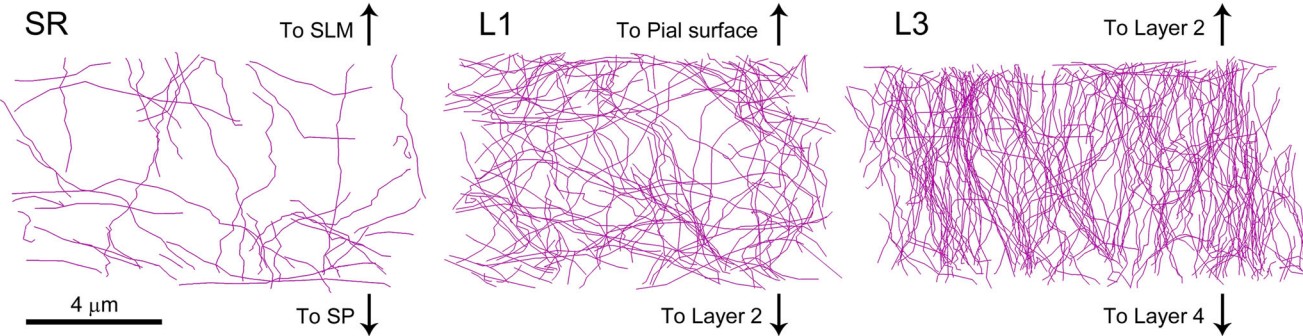

**Fig. 8 | Non-synaptic fibers.** Fibers traversing the stacks of serial sections without establishing any synapses (purple) are represented in the stratum radiatum of the hippocampus (SR) and layers 1 and 3 of the somatosensory cortex (L1 and L3, respectively). The frontal faces of the stacks are oriented towards the observer; mesial is to the left, and lateral to the right. The top-down orientation of each stack is represented by the corresponding arrows (SLM stratum lacunosum-moleculare, SP stratum pyramidale). The fibers were traced from stacks of 305 (SR and L1) and 263 serial sections (L3). The samples show different densities of fibers (see also Table 1 and Fig. 9b). Fibers in the SR and L3 run in all directions, while the course of fibers in L3 seems to be mostly vertical.

inhibitory synapses is similar to the mouse data, including the fact that layer 1 has a higher percentage of inhibitory synapses than layer 3[54,55].

As mentioned above, our study shows that classification criteria such as whether the postsynaptic target is a dendritic spine or shaft are not adequate to distinguish between excitatory and inhibitory synapses. For example, the fact that inhibitory synapses prefer dendritic shafts does not imply that any synapse on the shaft of a dendrite is inhibitory. On the contrary, in the present study we show that most synapses on the shaft of dendrites are in fact excitatory. Similarly, dendritic spines usually establish excitatory synapses, but they can also establish inhibitory synapses, usually accompanied by another excitatory synapse on the same spine[56–59]. The proportion of inhibitory synapses that are established on dendritic spines seems to be very variable. Our present results indicate that 23% to 40% of all inhibitory synapses are axo-spinous in the three regions studied. The percentages reported in other studies in rodents range from 17% to 40%[55–57,60–62]. Leaving aside region, species and age differences between studies, this variability may also be increased by the fact that inhibitory axo-spinous synapses are a very dynamic subpopulation, given that they are removed and re-established very frequently[63].

Another significant contribution of the present research lies in our ability to interpret synaptic density (measured as synapses per cubic micron) in relation to the number of axons (quantified by the total axonal length per cubic millimeter) and the corresponding count of synapses established by these axons (measured as the linear synaptic density). It should be stressed that the determination of linear synaptic densities was possible because the lengths of dendrites and axons were actually measured in 3D from their respective skeletons. In this way, we can take accurate measurements of the course of fibers, which would have been impossible to obtain from 2D images. For example, if we compare SR and L1, it is clear that excitatory synapses in the SR are much more numerous. This is mainly because excitatory axons in the SR establish many more synapses per micron than axons in L1, while the total lengths of axons in both regions are similar. A different scenario is seen when we compare L1 and L3; the overall number of excitatory synapses in L1 is higher than in L3, but in this case the difference is

mainly due to the higher total length of excitatory axons in L1, since the linear densities of synapses are similar. Likewise, inhibitory synapses in the SR are scarce, mainly because there are less inhibitory axons there than in L1 or L3. Therefore, we not only provide the overall quantity of synapses in a given region, but also a detailed quantitative interpretation of the factors on which this value depends.

Dendrites with spines clearly predominate over smooth dendrites in our samples. In the SR of CA1, the vast majority of dendrites originate from pyramidal cells, whose cell bodies form the pyramidal cell layer. In L1 and L3, we find dendrites that originate from the apical dendrites of neurons located in deeper layers. Furthermore, In L3, the basal and apical dendritic arborizations of local pyramidal cells contribute to the dendritic population. Therefore, our samples are a mixture of dendrites of different branching orders and thicknesses originating from different cell types. It has been shown, both in the hippocampus[49] and the neocortex[64,65], that synaptic and spine densities of dendrites vary with branching order and distance to the soma. It is highly likely that this contributes to the variability of the linear densities of synapses on dendrites that we have found in the three regions. However, further research is needed to rule out the influence of other factors such as dendrite thickness or local variability, which have not been addressed in the present study.

An interesting finding regarding spiny dendrites is that the density of synapses on spines does not correlate with the density of synapses on the dendritic shaft. In other words, the number of axo-spinous synapses in a given dendrite is independent of the number of synapses on the dendritic shaft. This was observed in the three regions studied. This finding reinforces the notion that dendritic spines are separate compartments and their synapses are regulated independently of those established on the dendritic shaft[62,66,67].

Dendrites without spines (smooth dendrites) were always scarce in our samples. In the SR, they most probably belong to any of the inhibitory interneuronal types present in the hippocampus, except horizontal trilaminar cells and O-LM cells, whose dendritic trees are restricted to the Stratum oriens[32,68]. In the neocortex, smooth dendrites may belong to an

https://doi.org/10.1038/s42003-024-06491-0                                              **Article**

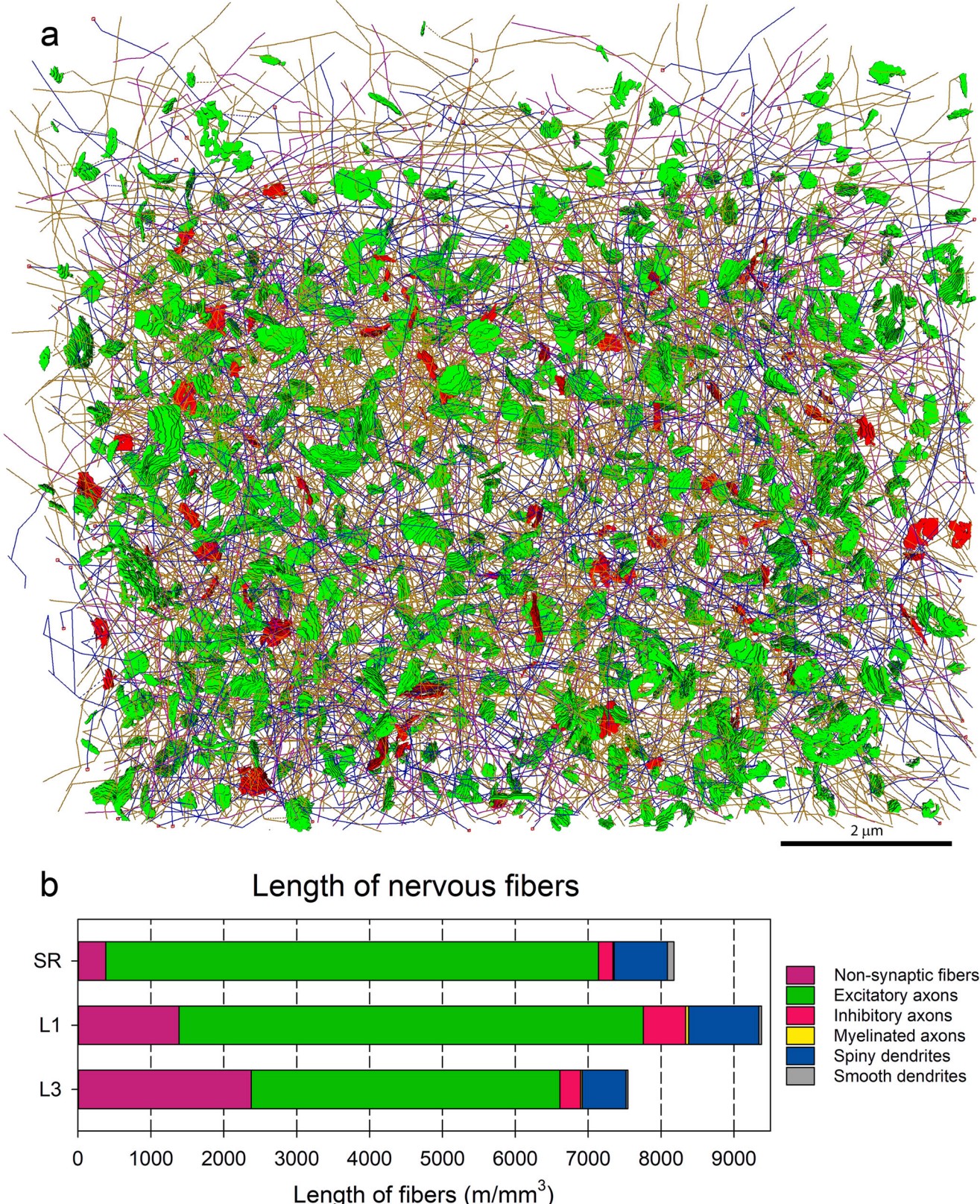

**Fig. 9 | Length of neuronal processes. a** The axons (orange), dendrites (blue), non-synaptic fibers (purple), as well as excitatory (green) and inhibitory synapses (red) connecting them, have been traced in a stack of serial images. This particular example was obtained from layer 1 of the mouse somatosensory cortex. Once all fibers have been traced, their respective lengths can be measured. **b** Lengths of the different types of nerve fibers that are present in stacks of serial images acquired from the stratum radiatum of the hippocampus (SR) and layers 1 and 3 of the somatosensory cortex (L1 and L3, respectively). Lengths are expressed as meters of fiber per cubic millimeter of tissue. See also Table 1.

even larger population of different inhibitory interneurons[27]. We consistently found that the mean densities of synapses on smooth dendrites were higher than on the shafts of spiny dendrites. This also indicates a different regulation of synaptic numbers, in this case between inhibitory interneurons, which usually have few or no spines, and the shafts of excitatory pyramidal cells.

Excitatory axons were, by far, the most common fiber type. In fact, the sum of excitatory axonal segments was six to eight times larger than spiny dendrites. Excitatory axons show heterogeneous linear densities of synapses, especially in the SR. Inhibitory axons were scarce in the three regions studied and they had similar linear densities of synapses in spite of their different origins. Interestingly, we found numerous axons that established repeated contacts with the same dendrite as well as numerous dendrites establishing multiple contacts with one or more axons. This discovery is meaningful as it has been suggested that the activity of a single spine can influence the plasticity of nearby spines through the shared exchange of proteins related to plasticity—or through the activation of synchronized synaptic inputs. These changes occur over various time scales and often lead to the spatial arrangement of spines into clusters or groups. This phenomenon, referred to as the "clustered plasticity hypothesis"[69–72], implies that spines do not function as individual units but rather form part of an intricate network that organizes them into groups, optimizing the connectivity patterns between dendrites and the surrounding axons.

To sum up, we have found four common traits in these different regions. First, we confirm that the most frequent synapse type is excitatory axo-spinous, followed by excitatory synapses on dendritic shafts, inhibitory synapses on shafts and inhibitory axo-spinous synapses. Second, linear densities of axo-spinous synapses were always the highest, followed by synapses on smooth dendrites, and synapses on the shafts of spiny dendrites. Third, the number of synapses established by individual axons and dendrites is far from homogeneous. Fourth, different types of nerve cell processes were heavily packed and intermingled, 7.5 to more than 9 km per cubic millimeter. Leaving aside non-synaptic fibers, excitatory axons were always the most common type, followed by spiny dendrites, inhibitory axons and smooth dendrites. Even though these patterns were present in the three locations studied, we have found quantitative differences that would have gone unnoticed in a qualitative analysis. Among the most salient quantitative differences are the high linear density of synapses established by excitatory axons in the SR and the relatively high number of inhibitory axons in L1.

However, our present results should be considered with caution, since they represent a single snapshot of three different regions and do not account for the local or interindividual variability described in previous studies[48,54,55]. A comprehensive study of any region, or a comparison between different regions, would require the analysis of multiple volumes of brain tissue. Our current methodological approach facilitates this process by offering detailed quantitative information in a robust and reliable way. This approach can offer a better understanding of the structure and dynamics of the brain, as well as the opportunity to explore how these aspects change during both normal and pathological conditions.

Finally, the strategy we propose for quantitatively examining synaptic connectivity in different brain regions has the potential to be particularly useful and reliable compared to other connectomics studies. The key strength of our method is its simplicity, since we leverage a schematic representation of neuronal processes and their connections. This approach significantly reduces the time and computational resources required compared to dense reconstructions, while still providing valuable quantitative data about the connectivity within a specific region. A further benefit of our approach is its compatibility with other methods. This implies that information about the caliber of fibers or the volume fractions occupied by different tissue components, which are not directly available from the skeletons, can still be obtained through other means, including dense reconstructions.

## Materials and methods

### Tissue preparation

For this study, three male C57 mice were sacrificed at postnatal week 8. All animals were handled in accordance with the guidelines for animal research set out in European Community Directive 2010/63/EU, and all procedures were approved by the local ethics committee of the Spanish National Research Council (CSIC). The animals were administered a lethal intraperitoneal injection of sodium pentobarbital (40 mg/kg) and were intracardially perfused with 4% paraformaldehyde in 0.12 M phosphate buffer (PB). The brain was then extracted from the skull and processed for electron microscopy. Briefly, the brains were extracted from the skull and post-fixed at 4 °C overnight in a solution composed of 2% paraformaldehyde, 2.5% glutaraldehyde and 0.003% $CaCl_2$ in 0.1 M cacodylate buffer. They were then washed in PB and 150 μm-thick vibratome sections were obtained.

Sections containing either the primary somatosensory cortex (hindlimb representation) or the dorsal hippocampus were selected with the help of an atlas[73]. Selected sections were osmicated for 1 h at room temperature in 1% $OsO_4$, 0.003% $CaCl_2$, and 0.1% FeCN in 0.1 M cacodylate buffer, and for another hour in the same solution without FeCN. After washing in PB, the sections were stained for 30 min with 1% uranyl acetate in 50% ethanol at 37 °C, and they were then dehydrated and flat-embedded in Araldite[74]. Embedded sections were glued onto blank Araldite stubs and trimmed. To select the exact location of the samples, we first obtained plastic semithin sections (1–2 μm thick) from the block surface and stained them with toluidine blue to identify cortical layers. These sections were then photographed with a light microscope. The last of these light microscope images (corresponding to the section immediately adjacent to the block face) was then collated with SEM photographs of the surface of the block. In this way, it was possible to accurately identify the regions of interest.

Brain tissue shrinks during processing for electron microscopy, especially during osmication and embedding. To estimate the shrinkage in our samples we measured the surface area and thickness of the vibratome sections with Stereo Investigator (MBF Bioscience, Williston, VT, USA), before and after processing[75]. Correction factors for volume, surface and linear measurements were 0.87, 0.91 and 0.96, respectively. All measurements have been corrected for tissue shrinkage according to these correction factors.

### Volume electron microscopy

We obtained stacks of serial images from layers I and III of the hindlimb representation of the somatosensory cortex (L1 and L3, respectively), and from the stratum radiatum (SR) of hippocampal CA1 area using a Crossbeam 540 electron microscope (Carl Zeiss NTS GmbH). This instrument combines a focused gallium ion beam (FIB) and a field emission scanning electron microscope (SEM) that can be used sequentially to acquire long series of high-resolution images. The FIB is used to mill the sample surface, removing a thin layer of material (20 nm thick). Then, the milling process is paused, and the freshly exposed surface is imaged with the SEM, using the in-column energy-selective backscattered electron detector. The milling and imaging cycles are repeated automatically, thus obtaining a stack of serial images that represents a three-dimensional sample of the tissue[11].

The stacks of images were taken from the neuropil, which is composed of axons, dendrites and glial processes, so the samples did not contain any cell somata, proximal dendrites in the immediate vicinity of the soma, or blood vessels. We used a 700 pA gallium current for the FIB milling process and 1.8 kV accelerating voltage for the SEM beam. The probe current of the SEM column was 1.80 nA. Dwell time was 3.33 μs per pixel. Acquisition time of each frame was 10.48 s. Noise reduction was performed by frame integration of 10 frames, so the total acquisition time of each final image was 1.75 min. Image resolution in the XY plane was 5 nm/pixel. Resolution in the Z axis (section thickness) was 20 nm, and frame size was 2048 × 1536 pixels. The numbers of serial sections per stack were 305, 305, and 263 for the SR, L1 and L3, respectively, corresponding to tissue volumes—after shrinkage correction—of 549 μm³ in the SR and L1 and 473 μm³ in L3.

## Identification, segmentation and quantification of synapses

Serial images were aligned (registered) with Fiji software[76] (https://fiji.sc).We used the "Register Virtual Stack Slices" plug-in, with a protocol that only allows translation of the images, with no rotation or deformation. The stack of images was then pre-processed with a Gaussian blur filter to eliminate noisy pixels. Then, synaptic junctions within the tissue were visualized and segmented in 3D with Espina software[23] (https://cajalbbp.csic.es/espina-2/). The segmentation algorithm applies a gray-level threshold to extract the voxels of the pre- and postsynaptic densities that appear as dark, electron-dense structures under the electron microscope[77]. As previously described[11,24,78], synaptic junctions with a prominent or thin postsynaptic density were classified as asymmetric or symmetric synaptic junctions, respectively[25,26]. The density of synapses was calculated by directly counting the number of segmented synapses inside a three-dimensional counting frame of known volume[11]. We used an unbiased stereological counting frame consisting of a 3D box spanning most of the available tissue volume and bounded by three acceptance planes and three exclusion planes[79]. Espina software counts the number of segmentations inside the box, including those touching the acceptance planes, and discarding those touching the exclusion planes[23].

## Skeletonized reconstruction of dendrites and axons

In order to describe the connectivity of the tissue, we used a tracing tool available in Espina software. With this tool, the user can navigate the stack and trace any nerve fiber by following its trajectory with the mouse (Fig. 1 and Supplementary Movie 1). A dendrite is identified as such when it is postsynaptic to any of the previously segmented synapses. Dendritic spines (spines, for simplicity) are also traced and connected to the shaft. The synapses established by each dendrite are associated to it at their corresponding locations (dendritic shaft or spines). Similarly, fibers are identified as axons when they are presynaptic with respect to any of the segmented synapses. Synapses are also associated with axons at the corresponding locations. In this way, we obtain a simplified version or skeleton of each fiber that includes its connectivity with other fibers. We also traced nerve fibers that did not establish any synaptic contact within the volumes studied. Given the absence of synapses, their classification as axons or dendrites could not be established with certainty and, thus, they were tagged as "non-synaptic fibers".

The software provides quantitative data regarding each skeleton. Users can select their desired information from a dialog box and this may vary depending on the type of segmentation. For example, available information for axons includes shaft length, number of synapses, and number of dendrites contacted, among other parameters. For dendrites, the number of dendritic spines is also provided, along with the number of synapses established on the shaft and on the spines (Fig. 1). To calculate the linear density of synapses, expressed as synapses per micron of shaft, we used dendritic and axonal skeletons that were at least 3 microns long.

## Statistics and reproducibility

Statistical analysis was performed with Graphpad Prism and IBM SPSS software packages. Given that homoscedasticity (Levene's test) and normality criteria (Kolmogorov-Smirnov and Shapiro-Wilk tests) were not met, we used nonparametric tests to compare groups. For multiple groups, we used the Kruskal-Wallis test (KW), with Bonferroni correction for pairwise comparisons. To compare two groups we used the Mann-Whitney $U$ test (MW). Contingency tables were analyzed with Chi-square tests.

## Data availability

Data generated or analyzed during this study are included in the main text, tables, figures and Supplementary Information. Numerical source data behind the graphs in the paper can be found in Supplementary Data 1. Any additional data are available from the corresponding author on request.

## Code availability

Espina software is available at https://cajalbbp.csic.es/espina-2/.

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

## Acknowledgements
We would like to thank Ana I. Garcia for technical assistance and Nick Guthrie for his excellent editorial assistance. This work was supported by the following Grants: PID2021-127924NB-I00 funded by MCIN/AEI/10.13039/501100011033; CSIC Interdisciplinary Thematic Platform - Cajal Blue Brain (PTI-BLUEBRAIN; Spain); CIBERNED, ISCIII, CB06/05/0066 and; the European Union's Horizon 2020 Framework Programme for Research and Innovation under Specific Grant Agreements No. 785907 (Human Brain Project SGA2) and No. 945539 (Human Brain Project SGA3).

## Author contributions
M.T.L., A.S., J.D.F. and A.M.P. designed research. F.P., M.T.L., A.S. and A.M.P. designed software. F.P. developed software. M.T.L. and A.S. tested and validated software. M.T.L., A.S. and J.R.R. performed research. M.T.L. and A.M.P. analyzed data. M.T.L., J.D.F. and AMP drafted the manuscript; all the authors reviewed the manuscript.

## Competing interests
The authors declare no competing interests.
