## [Peer Review File · Communications Biology]

Reviewers' comments:

Reviewer #1 (Remarks to the Author):

In this paper, Turegano-Lopez et al. present an analysis of volume electron microscopy datasets of brain samples. They image by FIB-SEM small regions in three different areas of the mouse brain. Using an update of a previously published software tool (Morales et al, 2011), they extract data about connectivity, length of the fibers, identity of the synaptic contacts and compare the brain regions with respect to these parameters.

The analysis tools are interesting and well presented, although not conceptually new. My main concern with this work is the low amount of data analyzed. The FIB-SEM stacks are very small to be representative of a brain area. Moreover, and perhaps more importantly, to be able to extract any meaningful information, the analysis of a single volume from a single mouse is not enough. The authors criticize in the introduction the limitations of dense reconstructions of large volumes for connectomics, because of the low throughput of such studies. They propose a different approach, which should give the possibility to analyze more samples, but then they present a comparison based on a single small stack for each brain region. It is a pity, because the authors have proven to be able to extract very detailed parameters from the data. Such analyses have the potential to provide interesting information, if done in a more statistically relevant way. My main comment is that the authors should acquire and analyze more datasets.

If time and/or capacity will not allow the authors to repeat the investigation for all 3 regions, my suggestions would be to further their analysis in a single brain region and acquire multiple FIB-SEM stacks in at least 3 animals. Looking at the intra- and inter-individual variability in a single region would be, in my opinion, extremely valuable information for the community.

More detailed comments:

- 1) Line 50: "we have traced all nerve fibers present within different brain samples": It sounds like they have solved the connectomics of several brains. Please rephrase this.
- 2) Line 130. Please add references about volume EM.
- 3) Line 136-138: "in order to examine multiple brain samples ... we believe that volume electron microscopy can be utilized more effectively" This is a strong statement. I would say that there can be complementary approaches. Also, to be fair, the authors have not analyzed multiple samples anyway...
- 4) Line 272: "Any fiber that was presynaptic to at least one synapse was tagged as an axon". I can imagine that, given the size of the FIB-SEM volume, there will be many axons that do not display any synaptic contact. How are these considered?
- 5) Line 472-475: "It could be argued that our sample is a mixture of dendrites of different branching orders and thickness, but the same argument could be applied to L1 and L3, ...". The volumes acquired are so small that there could be a bias in one or the other sample, if a sub-area with peculiar characteristics was accidentally selected. Instead of speculating, the authors should compare more volumes.
- 6) Line 572. SEM acquisition parameters should be better described. What was the SEM current used? Pixel dwell time?
- 7) Line 578. Which Fiji plug-in has been used for the alignment?
- 8) Line 587. "an unbiased stereological counting frame". Why do you need a stereological approach? Can't all the synapses be counted, if they are segmented. Please explain this point.
- 9) Line 618-621. I hope the authors would agree to share their FIB-SEM datasets in a public repository.

Reviewer #2 (Remarks to the Author):

In the work "Nanoconnectivity: a quantitative approach to examine the structural complexity of the

brain", Turegano-Lopez et al. present a thorough analysis of the dense composition of 3 selected volumes from the mouse hippocampus and primary sensory cortex. Results display the connectivity patterns, the relative composition of excitatory and inhibitory synapses, the nature of the neurites (axons/vs dendrites), the distribution and diversity of axodendritic connections and also, which is in my opinion one of the most compelling information, data related to the linear density of synapses along dendrites.

The fact that the work is done on data acquired by volume SEM, namely FIB-SEM, comes with impactful numbers and findings that can only be achieved by such techniques (e.g. linear density), but also suffers from inherent weakness of the technique, i.e. its low throughput. Inevitably, such imaging technique limit the possibility to perform biological replicate (different mice) and inter-sample replicate (multiple ROIs from the same sample).

Whilst it should not affect the relevance of the method, my major advice on the paper would be to clearly discuss this point and propose ways to mitigate the methodological bottlenecks, and potentially propose alternative methods.

Nevertheless, the paper is nicely written and supported by impeccable figures, accurate literature and thorough data analysis.

The methodology, which is one strong focus of the work, is well described and would be easily reproduced by other laboratories. The instrumentation utilized in the study start to be well spread throughout the community, and the software tools to perform the analyses are available. For this reason, I believe the work is of interest to the field of neuroscience.

Some specific comments:

- On the method to automatically segment synapses: it is based on thresholding, which is fine since these structures do stand out in the dataset. Yet, I would expect that such method would generate false positive in the dataset. Is it the case? Could the authors describe how they have evaluated such occurrences?

- Line 199: the authors mention an "unbiased 3D counting frame". Taking only one frame is dangerous. What if they fall in a richer / poorer region? It would be very interesting to use the same dataset and apply counting methods from stereology. They could estimate the volume density by applying the disector method for example (for reference, see .Lucocq, J. M. & Hacker, C. Cutting a fine figure: On the use of thin sections in electron microscopy to quantify autophagy. *Autophagy* 9, 1443–1448 (2013). The authors should also estimate the heterogeneity of the whole cube. How was this cube selected to start with. Is it an unbiased way of sampling SR / L1 etc ... Did they compare several mice in the similar region. When reading stereology papers, applying strict counting rules, unbiased approaches, impactful conclusions can be reached with an optimised quantity of data, which is precisely one of the goal of this study.

- Line 263: Linear densities of synapses: In my opinion this metric is really interesting with respect to justifying the use of vEM to build an understanding of the connectivity within a brain region. Indeed, absolute synapse densities could be computed from 2D slices (see my previous comment re stereology), but defining a smooth dendrite vs spiny one, and following the distribution of synapses along their length can only be achieved in volumes. This should be stressed more in the result, but also in the discussion.

- Line 542: methods section. This sentence is ambiguous as one can't know if the osmium step was done in PB or in Caco. Please clarify.

- Line 555: Evaluation of tissue deformation by sample prep has also been discussed in a more recent paper that would be worth considering. As this paper mentions brain tissue, it would be worth commenting on potential discrepancies. Ströh, S., Hammerschmith, E. W., Tank, D. W., Seung, H. S. & Wanner, A. A. In situ X-ray-assisted electron microscopy staining for large biological samples. *Elife* 11,

e72147 (2022).

-

Reviewer #3 (Remarks to the Author):

In the paper Turegano- Lopez et al, the authors provided results from a descriptive quantitative ultrastructural 3D analysis of 3 brain regions (stratum radiatum of the hippocampus and layers 1 and 3 of the primary somatosensory cortex). The authors implemented a new analysis in the Espina software. The main findings of the paper related to their reported enhanced image resolution (5x5x20 nm³) and their unequivocal ability to distinguish bonafide synapses, compared to other techniques and previous publications. The authors showed quantitative (not qualitative) changes in the density of synapses, the differential proportion of excitatory and inhibitory synapses, the location of synapse formation with respect to dendrites (axo-spinous or axo-shafts),... between 3 different brain areas.

My only negative comment relates to the purely descriptive nature of the paper. This potent analysis tool could have been used to compare ultrastructural changes in the brains of animals of different sexes, ages, physiological states (pregnancy...), or disease conditions.

Apart from that comment, I only have medium/minor that should be addressed:

Comments on the Results section:

- In Fig. 1, the brain region should be mentioned in the figure legend. Figure 1b is blurry and difficult to read. Not all the analysed parameters mentioned in lines 156-160 are shown in Fig. 1b. I recommend modifying the size of Fig. 1b to include all the parameters that the software can analyse (all the columns). I suggest adding an additional image containing the remaining columns.
- It should be more clearly stated in the manuscript that AS synapses are considered excitatory and SS inhibitory. I would state this around lines 191-192, and refer to the references provided to justify it. This statement is clearly provided in lines 271-274, but lacking this information earlier in the manuscript can be confusing. For example, Fig. 2n refers to the % of AS and SS in different brain regions, but it is stated as excitatory/inhibitory synapses, without mentioning the AS/SS classification. I would also cite Suppl. Table 1 in Fig. 2 figure legend.
- The results presented in lines 235-244, referring to Table S3, should also state that they refer to Fig. 3f.
- The analysis of the heterogeneously linear densities of synapses in inhibitory synapses could also be included as a supplementary figure.
- There is no table showing the data presented in lines 339-347.
- Line 135: Typo lacking “)” (see, for example (13, 18, 19)).
- First time using the term FIB-SEM in line 142. Please define here the term FIB-SEM, instead of in Fig. 2 figure legend.
- Line 212: Sentence needs rewriting.
- Line 231: The number 10.02% should be 10.27%, as stated in Table S7.

Comments on the Statistical methods:

- Did the authors use any test to analyse the normality distribution of their data?
- Only 1 SR, 1 L1 and 1 L3 region were used per animal?
- Values in columns should be represented as the mean \pm SE.
- Individual values in the columns should be presented as dot points.
- Additional statistical information is lacking in each figure legend, such as how many regions were analysed per condition and from how many animals.

Comments to the Discussion section:

-In the discussion, regarding AS (excitatory) and SS (inhibitory), I recommend including a comparison with other studies already published.

-Include in the discussion that the results have been obtained from male specimens (8 weeks old), and briefly discuss differences between different sexes and ages.

-Discuss the origin and function of smooth dendrites, in comparison to spiny dendrites.

-Discuss whether the non-homogeneous distribution of linear densities of synapses within individual dendrites (Fig. 4), can be associated with the order of dendrites (level of dendrite arborisation). The statement presented in lines 472-477 of the discussion is not very convincing. Could the authors correlate the densities of synapses in dendrites with, for example, the thickness of the dendrites?

Response to referees

Reviewers' comments:

Reviewer #1 (Remarks to the Author):

In this paper, Turegano-Lopez et al. present an analysis of volume electron microscopy datasets of brain samples. They image by FIB-SEM small regions in three different areas of the mouse brain. Using an update of a previously published software tool (Morales et al, 2011), they extract data about connectivity, length of the fibers, identity of the synaptic contacts and compare the brain regions with respect to these parameters.

The analysis tools are interesting and well presented, although not conceptually new. My main concern with this work is the low amount of data analyzed. The FIB-SEM stacks are very small to be representative of a brain area. Moreover, and perhaps more importantly, to be able to extract any meaningful information, the analysis of a single volume from a single mouse is not enough. The authors criticize in the introduction the limitations of dense reconstructions of large volumes for connectomics, because of the low throughput of such studies. They propose a different approach, which should give the possibility to analyze more samples, but then they present a comparison based on a single small stack for each brain region. It is a pity, because the authors have proven to be able to extract very detailed parameters from the data. Such analyses have the potential to provide interesting information, if done in a more statistically relevant way. My main comment is that the authors should acquire and analyze more datasets.

If time and/or capacity will not allow the authors to repeat the investigation for all 3 regions, my suggestions would be to further their analysis in a single brain region and acquire multiple FIB-SEM stacks in at least 3 animals. Looking at the intra- and inter-individual variability in a single region would be, in my opinion, extremely valuable information for the community.

Response:

We acknowledge the variability inherent in brain tissue samples, and we concur with the reviewer's assessment that additional samples are necessary for a comprehensive comparison between the hippocampus and the somatosensory cortex. However, it is crucial to clarify that the primary objective of our study was not to complete the comparison between these brain regions. Instead, our focus was on the development and validation of a tool that enables us to conduct such comparisons on a reliable quantitative basis. The three samples we used were selected to maximize diversity, as outlined in the Introduction. Our aim was to ascertain whether discernible differences could be identified. We believe that the results affirmatively support this hypothesis, as our methodological approach yields very detailed connectivity parameters. Consequently, our approach

emerges as a potent tool that can facilitate comparisons between regions, layers, individuals, diseases or species, provided that the statistical design is adequate.

Furthermore, we respectfully disagree with the assertion that we have analyzed a low amount of data. In fact, our study characterizes over 150 dendrites, more than 1100 axons, and more than 3000 synapses. We have been able to measure very precisely every single dendrite and axon, with their corresponding synaptic connections. In all sincerity, we believe that this substantial volume of data not only validates our methodological approach, but also holds intrinsic value in and of itself.

We reiterate that our aim was not to complete the comparison between regions, but rather to develop and validate a methodological approach that is both feasible and suitable for the task. Extending our study to two more animals would necessitate at least doubling the amount of work and time required to complete the study. Even if completion of that research were feasible within a reasonable timeframe, the outcome would remain incomplete because only one layer of the hippocampus would have been compared with two layers of the somatosensory cortex. Moreover, focusing solely on a single region, as suggested by the reviewer, would fail to provide a comprehensive understanding of either the hippocampus or the somatosensory cortex on a broader scale.

In fact, we are currently studying the structure of the six layers of the somatosensory cortex with the methodology described in the present paper. This undertaking entails analyzing at least 21 stacks from three animals, with separate analysis of layers Va and Vb. However, completion of this work remains several months away, and we plan to publish the findings in a separate paper.

To clarify this issue, we have added the following sentences to the Discussion section (lines 541-546):

“However, our present results should be considered with caution, since they represent a single snapshot of three different regions and do not account for the local or interindividual variability described in previous studies (48, 54, 55). A comprehensive study of any region, or a comparison between different regions, would require the analysis of multiple volumes. Our current methodological approach facilitates this process by offering detailed quantitative information in a robust and reliable way.”

More detailed comments:

1) Line 50: “we have traced all nerve fibers present within different brain samples”: It sounds like they have solved the connectomics of several brains. Please rephrase this.

We agree with the suggestion of the reviewer. In the Abstract, we have rephrased the sentence:

“Using volume electron microscopy and dedicated software we have traced all nerve fibers present within different brain samples”,

and it now reads (lines 49-50):

“We have traced all nerve fibers present within different samples of brain tissue using volume electron microscopy and dedicated software”

2) Line 130. Please add references about volume EM.

We have added these references at the location in the text suggested by the reviewer:

18. Kievits, A. J., Lane, R., Carroll, E. C. & Hoogenboom, J. P. How innovations in methodology offer new prospects for volume electron microscopy. *J Microsc* **287**, 114–137 (2022).

19. McCafferty, C. L. *et al.* Integrating cellular electron microscopy with multimodal data to explore biology across space and time. *Cell* **187**, 563–584 (2024).

20. Collinson, L. M. *et al.* Volume EM: a quiet revolution takes shape. *Nat Methods* **20**, 777–782 (2023).

3) Line 136-138: “in order to examine multiple brain samples ... we believe that volume electron microscopy can be utilized more effectively” This is a strong statement. I would say that there can be complementary approaches. Also, to be fair, the authors have not analyzed multiple samples anyway...

We agree that the sentence might be confusing, so we have rewritten it according to the suggestion of the reviewer. The following sentence:

“However, in order to examine multiple brain samples (from the same or different individuals), we believe that volume electron microscopy can be utilized more effectively by employing an alternative approach.”

has been rephrased, and now reads (lines 136-137):

“However, we believe that volume electron microscopy can be utilized more effectively by employing an alternative or complementary approach.”

4) Line 272: “Any fiber that was presynaptic to at least one synapse was tagged as an axon”. I can imagine that, given the size of the FIB-SEM volume, there will be many axons that do not display any synaptic contact. How are these considered?

We omitted neuronal processes that do not form synapses from our analysis because our primary focus was on connectivity. However, prompted by the reviewer’s comment, we

now recognize that quantifying these segments may have standalone significance. Nevertheless, given that these fibers do not establish synapses, categorizing them as axons or dendrites presents challenges, particularly for short segments traversing the periphery of the stack. Therefore, we have tagged them as “non-synaptic fibers”.

We have added the following information regarding these fibers to the manuscript:

In the Results section, lines 350-357, we have added the following paragraph:

“Non-synaptic fibers

We also traced and measured the fibers that traverse the volumes without establishing any synapses. Typically, these are thin, unbranched fibers resembling axonal segments. However, due to the absence of synapses, their classification as dendrites or axons cannot be established with certainty. Therefore, we categorized them separately as “non-synaptic” fibers. In the SR and L1, these fibers exhibited multidirectional trajectories and accounted for 4.67% and 14.79% of the total length of nerve fibers, respectively. In L3, they were more numerous (31.49% of the total length of fibers) and displayed a preference for vertical orientation (Table 1, Figure 8).”

In the Results section we have rewritten the following paragraph:

“The estimated total length of nerve processes (dendrites + axons) per cubic mm of neuropil was similar in the SR and L1 (7793.73 and 7991.50 meters/mm³, respectively), while the total length of fibers was lower in L3 (5170.00 meters/mm³) (Figure 8b; Table 1). Although there were differences between the three regions studied, there was a common pattern in the relative length of the different types of fibers. Excitatory axons were, by far, the most common type of fiber in all locations since they comprised between 79.73% and 86.75% of the total length of all nerve fibers (Table 1). The second most common population of fibers was composed of spiny dendrites, which accounted for 9.39% to 12.03% of the total number of fibers. Third, inhibitory axons, whose total length ranged from 2.50% to 7.24% of all fibers. Finally, smooth dendrites and myelinated axons were scarce in our samples, with percentages around or below 1% of the total length of nerve fibers (Table 1).”

The rewritten paragraphs read as follows (lines 370-382):

“The estimated total length of nerve fibers per cubic mm of neuropil ranged from 7546 meters/mm³ in L3, to 9378 meters/mm³ in L1 (Table 1; Figure 9b). Non-synaptic fibers were the most variable category, ranging from 4.67% to 31.49% of the total length of fibers. If we only consider the fibers that establish synapses and were unambiguously identified as dendrites and axons, the estimated total length was similar in the SR and L1 (7793.73 and 7991.50 meters/mm³, respectively), while the total length of these fibers was lower in L3 (5170.00 meters/mm³) (Table 1; Figure 9b).

Although there were differences between the three regions studied, there was a common pattern in the relative length of the different types of fibers, leaving aside non-synaptic

fibers. Excitatory axons were, by far, the most common type in all locations, comprising between 56.15% and 82.70% of the total length of all nerve fibers (Table 1). These were followed by spiny dendrites (which accounted for 7.97% to 10.25% of the total number of fibers) and inhibitory axons (2.39% to 6.17% of all fibers). Finally, smooth dendrites and myelinated axons were scarce in our samples, with percentages around or below 1% of the total length of nerve fibers (Table 1; Figure 9b).”

We have added a new figure for non-synaptic fibers (Figure 8)

The old Figure 8 has been renumbered to Figure 9. We have modified Figures 9a and 9b to include non-synaptic fibers.

We have also updated Table 1 to include non-synaptic fibers:

Region	Length of non-synaptic fibers (m/mm ³)	Length of excitatory axons (m/mm ³)	Length of inhibitory axons (m/mm ³)	Length of myelinated axons (m/mm ³)	Length of spiny dendrites (m/mm ³)	Length of smooth dendrites (m/mm ³)	Total length of nerve fibers (m/mm ³)
SR	382.06 4.67%	6761.19 82.70%	195.14 2.39%	17.72 0.22%	731.79 8.95%	87.89 1.08%	8175.79 100%
L1	1387.28 14.79%	6371.80 67.94%	578.53 6.17%	42.68 0.46%	961.27 10.25%	37.22 0.40%	9378.79 100%
L3	2376.22 31.49%	4237.03 56.15%	282.96 3.75%	21.16 0.28%	601.56 7.97%	27.30 0.36%	7546.22 100%

Table 1. Lengths of the different types of neuronal processes in the stratum radiatum (SR), and layers 1 and 3 of the somatosensory cortex (L1 and L3). Data are given as the length of fibers in meters per cubic mm, and the percentage of the total length of nerve fibers in each region.

In the Discussion, the sentence:

“Fourth, different types of nerve cell processes were heavily packed and intermingled, 5 to 8 kilometers per cubic micron, but excitatory axons were always the most common type, followed by spiny dendrites, inhibitory axons and smooth dendrites.”

has been corrected and now reads (lines 533-536):

“Fourth, different types of nerve cell processes were heavily packed and intermingled, 7.5 to more than 9 kilometers per cubic millimeter. Leaving aside non-synaptic fibers,

excitatory axons were always the most common type, followed by spiny dendrites, inhibitory axons and smooth dendrites.”

In Materials and Methods, we have added (lines 630-633):

We also traced nerve fibers that did not establish any synaptic contact within the volumes studied. Given the absence of synapses, their classification as axons or dendrites could not be established with certainty and, thus, they were tagged as “non-synaptic fibers”.

5) Line 472-475: “It could be argued that our sample is a mixture of dendrites of different branching orders and thickness, but the same argument could be applied to L1 and L3, ...”. The volumes acquired are so small that there could be a bias in one or the other sample, if a sub-area with peculiar characteristics was accidentally selected. Instead of speculating, the authors should compare more volumes.

We agree that further research is needed to solve this issue (please see our response to the general considerations), so we have modified our discussion of the topic as follows.

The paragraph:

“Dendrites with spines clearly predominate over smooth dendrites in our samples. In the SR of CA1, the vast majority of dendrites originate from the main shaft and collateral branches of apical dendrites of pyramidal cells, whose cell bodies form the pyramidal cell layer. However, in L1 and L3, the postsynaptic dendrites are very heterogeneous, since they comprise the terminal ramifications of apical dendrites of neurons located in deeper layers (in L1), plus the ramifications of basal and apical dendrites of local pyramidal cells (in L3). In spite of this apparent homogeneity of dendrites in the SR, the linear densities of synapses of dendrites in SR are more variable there than in L1 and L3, as indicated by a wider probability distribution. It could be argued that our sample is a mixture of dendrites of different branching orders and thicknesses, but the same argument could be applied to L1 and L3, which show a more homogeneous distribution of linear densities of synapses. Therefore, even if we assume that the linear density of synapses on dendrites depends on the branching order, thickness, or cell of origin of the dendrites, this dependency seems to be different in the SR and the neocortex.”

now reads (lines 486-496):

“Dendrites with spines clearly predominate over smooth dendrites in our samples. In the SR of CA1, the vast majority of dendrites originate from pyramidal cells, whose cell bodies form the pyramidal cell layer. In L1 and L3, we find dendrites that originate from the apical dendrites of neurons located in deeper layers. Furthermore, In L3, the basal and apical

dendritic arborizations of local pyramidal cells contribute to the dendritic population. Therefore, our samples are a mixture of dendrites of different branching orders and thicknesses originating from different cell types. It has been shown, both in the hippocampus (49) and the neocortex (64, 65), that synaptic and spine densities of dendrites vary with branching order and distance to the soma. It is highly likely that this contributes to the variability of the linear densities of synapses on dendrites that we have found in the three regions. However, further research is needed to rule out the influence of other factors such as dendrite thickness or local variability, which have not been addressed in the present study.”

49. Megías, M., Emri, Z., Freund, T. F. & Gulyás, A. I. Total number and distribution of inhibitory and excitatory synapses on hippocampal CA1 pyramidal cells. *Neuroscience* **102**, 527–540 (2001).

64. Larkman, A. U. Dendritic morphology of pyramidal neurones of the visual cortex of the rat: I. Branching patterns. *J Comp Neurol* **306**, 307–319 (1991).

65. Ballesteros-Yáñez, I., Benavides-Piccione, R., Elston, G. N., Yuste, R. & DeFelipe, J. Density and morphology of dendritic spines in mouse neocortex. *Neuroscience* **138**, 403–409 (2006).

6) Line 572. SEM acquisition parameters should be better described. What was the SEM current used? Pixel dwell time?

We have added the following information (lines 597-600):

The probe current of the SEM column was 1.80 nA. Dwell time was 3.33 μ s per pixel. Acquisition time of each frame was 10.48 s. Noise reduction was performed by frame integration of 10 frames, so the total acquisition time of each final image was 1.75 minutes.

7) Line 578. Which Fiji plug-in has been used for the alignment?

We have added this information at the location in the text suggested by the reviewer. The original sentence:

“Serial images were aligned (registered) with Fiji software (69) (<https://fiji.sc>), with a protocol that only allows translation of the images, with no rotation or deformation.”

has been rewritten as (lines 606-608):

“Serial images were aligned (registered) with Fiji software (76) (<https://fiji.sc>). We used the “Register Virtual Stack Slices” plug-in, with a protocol that only allows translation of the images, with no rotation or deformation.”

8) Line 587. “an unbiased stereological counting frame”. Why do you need a stereological approach? Can't all the synapses be counted, if they are segmented. Please explain this point.

There are three possible ways of counting the objects inside a (virtual) box. If you count all the objects inside the box, including those that traverse its faces, you are in fact overestimating the number of objects contained in the box. On the contrary, if you only count the objects that are inside the box but do not touch any of its faces, you are underestimating the number of objects. To avoid these biases, you can use an “unbiased stereological counting frame”. In 3D, this counting frame is a box in which three of its faces are considered as acceptance planes (objects touching these planes will be counted) and the other three faces as exclusion planes (objects touching these planes will not be counted). Even though the differences between the three counting methods may not be substantial, we have designed our software (23) to use an unbiased stereological counting frame, as described in reference (79). We tested this method experimentally in the past, showing its advantages with respect to the disector and size-frequency methods, so we have also added reference (11) at this location in the text.

To clarify this issue, we have rewritten the following sentence at the end of the section “Identification, segmentation and quantification of synapses”, in Materials and Methods.

The sentence:

“The density of synapses was calculated with Espina software by directly counting the number of segmented synapses inside an unbiased stereological counting frame of known volume.”

now reads (lines 614-619):

“The density of synapses was calculated by directly counting the number of segmented synapses inside a three-dimensional counting frame of known volume (11). We used an unbiased stereological counting frame consisting of a 3D box spanning most of the available tissue volume and bounded by three acceptance planes and three exclusion planes (79). Espina software counts the number of segmentations inside the box, including those touching the acceptance planes, and discarding those touching the exclusion planes (23).”

9) Line 618-621. I hope the authors would agree to share their FIB-SEM datasets in a public repository.

As indicated in the manuscript (lines 658-661), data sets and software are available from the corresponding author on request. We have also updated the link where Espina software can be downloaded:

<https://cajalbbp.csic.es/en/espina-2/>

Reviewer #2 (Remarks to the Author):

In the work “Nanoconnectivity: a quantitative approach to examine the structural complexity of the brain”, Turegano-Lopez et al. present a thorough analysis of the dense composition of 3 selected volumes from the mouse hippocampus and primary sensory cortex. Results display the connectivity patterns, the relative composition of excitatory and inhibitory synapses, the nature of the neurites (axons/vs dendrites), the distribution and diversity of axodendritic connections and also, which is in my opinion one of the most compelling information, data related to the linear density of synapses along dendrites.

The fact that the work is done on data acquired by volume SEM, namely FIB-SEM, comes with impactful numbers and findings that can only be achieved by such techniques (e.g. linear density), but also suffers from inherent weakness of the technique, i.e. its low throughput. Inevitably, such imaging technique limit the possibility to perform biological replicate (different mice) and inter-sample replicate (multiple ROIs from the same sample). Whilst it should not affect the relevance of the method, my major advice on the paper would be to clearly discuss this point and propose ways to mitigate the methodological bottlenecks, and potentially propose alternative methods.

Nevertheless, the paper is nicely written and supported by impeccable figures, accurate literature and thorough data analysis.

The methodology, which is one strong focus of the work, is well described and would be easily reproduced by other laboratories. The instrumentation utilized in the study start to be well spread throughout the community, and the software tools to perform the analyses are available. For this reason, I believe the work is of interest to the field of neuroscience.

Response:

We agree with the reviewer's suggestion that more samples should be analyzed to perform a comprehensive comparison between brain regions. However, it is essential to clarify that our primary objective was to design and develop a technique that would enable us to perform these comparisons on a solid and reliable quantitative basis. To test the efficacy of our approach, we selected three brain regions that are, a priori, very different in terms of connectivity. Our intention was to ascertain whether our methodology could accurately identify quantitative differences (and similarities) among these regions. We believe that our methodological approach fulfills our expectations, as we have been able to provide detailed quantitative data from each of the three regions.

It is evident that our current study does not address local or inter-individual variability, which undoubtedly necessitates the analysis of additional volumes. Currently, we are in

fact studying the six layers of the somatosensory cortex with the methodology described in the present paper. This involves tens of stacks of serial sections to analyze. This is a time-consuming task and the completion of this work is still months away. We plan to publish the results in a separate paper (please see also our response to reviewer #1 on this matter).

In any case, given our previous experience in addressing sample variability in earlier studies, we have added the following sentences to the Discussion (lines 541-546):

“However, our present results should be considered with caution, since they represent a single snapshot of three different regions and do not account for the local or interindividual variability described in previous studies (48, 54, 55). A comprehensive study of any region, or a comparison between different regions, would require the analysis of multiple volumes. Our current methodological approach facilitates this process by offering detailed quantitative information in a robust and reliable way.”

Some specific comments:

- On the method to automatically segment synapses: it is based on thresholding, which is fine since these structures do stand out in the dataset. Yet, I would expect that such method would generate false positive in the dataset. Is it the case? Could the authors describe how they have evaluated such occurrences?

False positives (non-synaptic objects that might be identified as actual synapses), typically do not present a significant issue in our samples. They usually involve membranes belonging to easily identifiable organelles such as mitochondria, multivesicular or multilamellar bodies, and they are readily disregarded. Additionally, certain intercellular junctions, such as gap junctions or adherens junctions, may occasionally resemble synaptic densities in some images. However, in this case, they can be readily distinguished by the absence of synaptic vesicles.

- Line 199: the authors mention an “unbiased 3D counting frame”. Taking only one frame is dangerous. What if they fall in a richer / poorer region? It would be very interesting to use the same dataset and apply counting methods from stereology. They could estimate the volume density by applying the disector method for example (for reference, see .Lucocq, J. M. & Hacker, C. Cutting a fine figure: On the use of thin sections in electron microscopy to quantify autophagy. *Autophagy* 9, 1443–1448 (2013). The authors should also estimate the heterogeneity of the whole cube. How was this cube selected to start with. Is it an unbiased way of sampling SR / L1 etc ... Did they compare several mice in the similar region. When reading stereology papers, applying strict counting rules, unbiased approaches, impactful conclusions can be reached with an optimised quantity of data, which is precisely one of the goal of this study.

The unbiased counting frame is in fact a 3D box that comprises most of the available tissue volume. It is “unbiased” since three of its faces are acceptance planes (segmentations touching them will be counted) and the remaining three faces are exclusion planes (any segmented object touching them will not be counted). To clarify this issue, we have rewritten the last sentence of the section “Identification, segmentation and quantification of synapses”, in Materials and Methods.

Regarding the disector, in a previous study we compared the disector method (and the size frequency method) with direct counting inside a 3D counting frame, highlighting the advantages of the latter (11). We have added this reference to the Methods section too.

Since Reviewer #1 raised a similar question, please also see our response to Reviewer #1, question 8.

The sentence:

“The density of synapses was calculated with Espina software by directly counting the number of segmented synapses inside an unbiased stereological counting frame of known volume.”

now reads (lines 614-619):

“The density of synapses was calculated by directly counting the number of segmented synapses inside a three-dimensional counting frame of known volume (11). We used an unbiased stereological counting frame consisting of a 3D box spanning most of the available tissue volume and bounded by three acceptance planes and three exclusion planes (79). Espina software counts the number of segmentations inside the box, including those touching the acceptance planes, and discarding those touching the exclusion planes (23).”

- Line 263: Linear densities of synapses: In my opinion this metric is really interesting with respect to justifying the use of vEM to build an understanding of the connectivity within a brain region. Indeed, absolute synapse densities could be computed from 2D slices (see my previous comment re stereology), but defining a smooth dendrite vs spiny one, and following the distribution of synapses along their length can only be achieved in volumes. This should be stressed more in the result, but also in the discussion.

In lines 472-475 we have added:

“It should be stressed that the determination of linear synaptic densities was possible because the lengths of dendrites and axons were actually measured in 3D from their respective skeletons. In this way, we can take accurate measurements of the course of fibers, which would have been impossible to obtain from 2D images.”

- Line 542: methods section. This sentence is ambiguous as one can't know if the osmium step was done in PB or in Caco. Please clarify.

We thank the reviewer for bringing this error to our attention. We have corrected the original sentence:

“Selected sections were osmicated for 1 hour at room temperature in PB with 1% OsO₄, 0.003% CaCl₂, and 0.1% FeCN in 0.1 M cacodylate buffer — and for another hour in the same solution without FeCN.”

and now it reads (lines 565-567):

“Selected sections were osmicated for 1 hour at room temperature in 1% OsO₄, 0.003% CaCl₂, and 0.1% FeCN in 0.1 M cacodylate buffer, and for another hour in the same solution without FeCN.”

- Line 555: Evaluation of tissue deformation by sample prep has also been discussed in a more recent paper that would be worth considering. As this paper mentions brain tissue, it would be worth commenting on potential discrepancies. Ströh, S., Hammerschmith, E. W., Tank, D. W., Seung, H. S. & Wanner, A. A. In situ X-ray-assisted electron microscopy staining for large biological samples. *Elife* 11, e72147 (2022).

It is well established that brain tissue undergoes shrinkage during processing for electron microscopy, and this shrinkage can vary depending on the methods employed. In the current article, we detailed how we estimate shrinkage in our samples. We believe that comparing these values with those obtained in other studies is beyond the scope of the present article, as it is contingent upon the specific methodologies employed.

Reviewer #3 (Remarks to the Author):

In the paper Turegano- Lopez et al, the authors provided results from a descriptive quantitative ultrastructural 3D analysis of 3 brain regions (stratum radiatum of the hippocampus and layers 1 and 3 of the primary somatosensory cortex). The authors implemented a new analysis in the Espina software. The main findings of the paper related to their reported enhanced image resolution (5x5x20 nm³) and their unequivocal ability to distinguish bonafide synapses, compared to other techniques and previous publications. The authors showed quantitative (not qualitative) changes in the density of synapses, the differential proportion of excitatory and inhibitory synapses, the location of synapse formation with respect to dendrites (axo-spinous or axo-shafts),... between 3 different brain areas.

My only negative comment relates to the purely descriptive nature of the paper. This potent analysis tool could have been used to compare ultrastructural changes in the brains of animals of different sexes, ages, physiological states (pregnancy...), or disease conditions.

Apart from that comment, I only have medium/minor that should be addressed:

Response:

We concur with the reviewer's comment that our paper is descriptive in nature. The primary objective of the study was to design and test a reliable tool and methodology for acquiring quantitative data from nervous tissue. We have thoroughly described, analyzed and discussed the connectivity data attainable through our approach. We do not perceive this descriptive aspect as negative; rather, we believe that our tool will prove instrumental in investigating changes occurring during both normal and pathological conditions, as commented in the last paragraph of the Discussion.

Comments on the Results section:

-In Fig. 1, the brain region should be mentioned in the figure legend. Figure 1b is blurry and difficult to read. Not all the analysed parameters mentioned in lines 156-160 are shown in Fig. 1b. I recommend modifying the size of Fig. 1b to include all the parameters that the software can analyse (all the columns). I suggest adding an additional image containing the remaining columns.

The list of parameters that Espina can provide is too long to fit in the figure, which is why we have selected only a few of them to show. For example, the list of spatial and geometric parameters that Espina provides for 3D segmentations includes volume (in voxels and in cubic nm), surface area, position (the x, y, z coordinates of the center of gravity or centroid

of the segmentation), the dimensions of the bounding box (the smaller rectangular prism that contains the 3D object), or the Feret's diameter (the diameter of the smaller sphere that circumscribes the object), among other items. In addition to these continuous variables, Espina stores categorical variables, such as the classification of a given synapse as asymmetric or symmetric, or its position on a dendritic spine or shaft. Furthermore, the information that can be extracted depends on the type of segmentation. For example, volume or surface make no sense for the skeletons of dendrites and axons; in this case, Espina provides the length, the number of spines, the number of synapses on spines and shafts, etc.

Since only a subset of these parameters may be relevant for a given analysis, the users can select the parameters they want in a dialog box. For example, if you only want to compare the total length of spiny and smooth dendrites, ticking the "Length" variable for both categories will be enough. However, if you also want to know how many synapses are established on dendrites, and how many of them are excitatory and inhibitory, you need to tick the corresponding variables too.

To clarify this point, we have tried to summarize this long explanation in the main text, as suggested by the reviewer:

In the Methods section, the sentence:

"The software provides quantitative information regarding each skeleton, including shaft length, number of synapses and, in the case of dendrites, the number of dendritic spines."

has been rewritten (lines 634-639), and now reads:

"The software provides quantitative information regarding each skeleton. Users can select their desired information from a dialog box and this may vary depending on the type of segmentation. For example, available information for axons includes shaft length, number of synapses, and number of dendrites contacted, among other parameters. For dendrites, the number of dendritic spines is also provided, along with the number of synapses established on the shaft and on the spines (Figure 1)."

Regarding the blurry appearance of the figure, we believe that this is a consequence of the relative low-resolution conversion to pdf. The original figure is much crispier, and we hope it will be reproduced accurately in the final version.

-It should be more clearly stated in the manuscript that AS synapses are considered excitatory and SS inhibitory. I would state this around lines 191-192, and refer to the references provided to justify it. This statement is clearly provided in lines 271-274, but lacking this information earlier in the manuscript can be confusing. For example, Fig. 2n refers to the % of AS and SS in different brain regions, but it is stated as excitatory/inhibitory synapses, without mentioning the AS/SS classification. I would also cite Suppl. Table 1 in Fig. 2 figure legend.

In lines 191-192, we state:

“In the cerebral cortex, AS are generally glutamatergic, excitatory synapses, while SS are mostly GABAergic, inhibitory synapses (27-29).”

To clarify this further, we have added the terms “excitatory” and “inhibitory” in the next sentence (Lines 192-195):

“It should be emphasized that the classification of any given synapse as AS (excitatory) or SS (inhibitory) was not based on the visualization of single sections. Rather, it relied on the examination of all serial sections in which each synaptic junction was visible (Figure 2a-h).”

In lines 273-275, we extended this classification to axonal segments:

“Axonal segments were further subclassified into excitatory, if they established AS; inhibitory, if they established SS; or myelinated axons, when they were covered by a myelin sheet.”

We have also cited the corresponding Tables in the respective figure legends.

-The results presented in lines 235-244, referring to Table S3, should also state that they refer to Fig. 3f.

We have corrected this omission, as suggested by the reviewer. We have added Figure 3f (line 237) to the sentence:

“The linear density of synapses was calculated as the number of synapses per micron of dendritic shaft (Figure 3f; Table S3).”

-The analysis of the heterogeneously linear densities of synapses in inhibitory synapses could also be included as a supplementary figure.

Frequency histograms similar to those shown in Figure 7b for excitatory axons could not be generated, as the numbers of inhibitory axons in our samples were very low, as indicated in Table S6.

-There is no table showing the data presented in lines 339-347.

We have added this table (Table S8) to the Supplementary material, as suggested by the reviewer.

-Line 135: Typo lacking “)” (see, for example (13, 18, 19)).

We have corrected the typo.

-First time using the term FIB-SEM in line 142. Please define here the term FIB-SEM, instead of in Fig. 2 figure legend.

We have defined FIB-SEM in line 142, as suggested by the reviewer:

“Briefly, we obtain stacks of serial images using focused ion beam milling and scanning electron microscopy (FIB-SEM) (10, 11).”

-Line 212: Sentence needs rewriting.

The sentence:

“Dendrites and axons were identified and traced after synaptic junctions had been segmented.”

has been rewritten (lines 213-214), and now reads:

“Once all synaptic junctions had been segmented, we proceeded to identify and trace dendrites and axons.”

-Line 231: The number 10.02% should be 10.27%, as stated in Table S7.

We thank the reviewer for bringing this error to our attention. We have corrected it in line 322.

Comments on the Statistical methods:

-Did the authors use any test to analyse the normality distribution of their data?

Yes, our data did not meet homoscedasticity (Levene’s test) and normality criteria (Kolmogorov-Smirnov and Shapiro-Wilk tests). Therefore, we opted to use non-parametric tests, as indicated in lines 643-645.

-Only 1 SR, 1 L1 and 1 L3 region were used per animal?

Yes, the stacks of serial sections that we used for the study are described in lines 594-603. Please also see our response to Reviewer #1’s general considerations.

-Values in columns should be represented as the mean \pm SE.

We have tried to give a coherent representation of data by using bar charts throughout the paper. Where appropriate (Fig. 3f, Fig 6d; Tables S3 and S6), we have presented the data as mean \pm standard error of the mean. This is indicated in the text and in the corresponding figures and tables. Furthermore, we have provided the value of n in the corresponding cases, ensuring that the information is comprehensive, and that the standard deviation can be calculated from these data, if needed. Other graphs, such as frequency histograms or the sum of lengths of nerve fibers, do not have an associated standard error of the mean or standard deviation. When presenting percentages, we have opted for stacked bar charts, as they facilitate comparisons. We also provide the absolute numbers from which the percentages are calculated, ensuring transparency and clarity in our data representation.

-Individual values in the columns should be presented as dot points.

Please see our response to the previous question.

-Additional statistical information is lacking in each figure legend, such as how many regions were analysed per condition and from how many animals.

The data from which each graph has been built are clearly indicated in the text and tables. We have added a reference to the corresponding tables in the figure legends.

Comments to the Discussion section:

-In the discussion, regarding AS (excitatory) and SS (inhibitory), I recommend including a comparison with other studies already published.

This topic is discussed in lines 443-467. We have added recent data from the Etruscan shrew

-Include in the discussion that the results have been obtained from male specimens (8 weeks

This is specified in Materials and Methods (Line 554)

-Discuss the origin and function of smooth dendrites, in comparison to spiny dendrites.

The origin and function of smooth dendrites are discussed in lines 505 to 512, where we propose that they probably belong to different types of inhibitory interneurons. The main difference is the relatively high linear density of synapses on smooth dendrites compared with the shaft of spiny dendrites, as indicated in the text.

-Discuss whether the non-homogeneous distribution of linear densities of synapses within individual dendrites (Fig. 4), can be associated with the order of dendrites (level of dendrite arborisation). The statement presented in lines 472-477 of the discussion is not very convincing. Could the authors correlate the densities of synapses in dendrites with, for example, the thickness of the dendrites?

We agree that these sentences may lack clarity (see also our response to reviewer #1, question 5). We have revised them as follows (lines 486-496):

“Dendrites with spines clearly predominate over smooth dendrites in our samples. In the SR of CA1, the vast majority of dendrites originate from pyramidal cells, whose cell bodies form the pyramidal cell layer. In L1 and L3, we find dendrites that originate from the apical dendrites of neurons located in deeper layers. Furthermore, In L3, the basal and apical

dendritic arborizations of local pyramidal cells contribute to the dendritic population. Therefore, our samples are a mixture of dendrites of different branching orders and thicknesses originating from different cell types. It has been shown, both in the hippocampus (49) and the neocortex (64, 65), that synaptic and spine densities of dendrites vary with branching order and distance to the soma. It is highly likely that this contributes to the variability of the linear densities of synapses on dendrites that we have found in the three regions. However, further research is needed to rule out the influence of other factors such as dendrite thickness or local variability, which have not been addressed in the present study.”

49. Megías, M., Emri, Z., Freund, T. F. & Gulyás, A. I. Total number and distribution of inhibitory and excitatory synapses on hippocampal CA1 pyramidal cells. *Neuroscience* **102**, 527–540 (2001).

64. Larkman, A. U. Dendritic morphology of pyramidal neurones of the visual cortex of the rat: I. Branching patterns. *J Comp Neurol* **306**, 307–319 (1991).

65. Ballesteros-Yáñez, I., Benavides-Piccione, R., Elston, G. N., Yuste, R. & DeFelipe, J. Density and morphology of dendritic spines in mouse neocortex. *Neuroscience* **138**, 403–409 (2006).

Reviewers' comments:

Reviewer #1 (Remarks to the Author):

I acknowledge that the authors consider their work as validation of their workflow, rather than an analysis of the differences of the connectivity and synapses of different brain regions. However, this is not clearly expressed in the manuscript, except a sentence in the discussion.

I agree that the validation of the workflow can be interesting per se, but the focus of the paper has to be changed, in my opinion.

1) If this is the aim of the manuscript, I think it should be clearly stated, starting from the abstract. In the current version of the paper, all the emphasis is still on the measurements, and the differences between brain regions. As the authors concurred, the volumes analyzed, are not significant for such a quantitative analysis.

2) As a proof-of-concept for a workflow, the authors should have an estimation of the accuracy of their approach, or at least discuss this point. I would also discuss more strengths and weaknesses of the approach.

On the other hand, I have appreciated the analysis of the non-synaptic fibers, which has improved the study, as well as the answers to all the other minor points I have raised in my previous report.

Reviewer #2 (Remarks to the Author):

I am satisfied by the new version of the manuscript. The authors carefully considered the questions that I asked and provided convincing answers.

I would have an additional comment and recommendation to make now that I have read through the paper once again:

One important message conveyed in the introduction is that this work provides a new and more efficient way to measure the connectome, diverging from the single-animal, exhaustive way of working that is taken by many in the field. I am missing a paragraph in the discussion that would come back to this. Maybe even the last paragraph. How does this new study address the difficulties of dense reconstructions? Where is the gain of the new technique? Is it faster? The speed of analysis, maybe also its computational load should be put in perspective of other methods.

Reviewer #3 (Remarks to the Author):

In the paper Turegano- Lopez et al, the authors provided results from a descriptive quantitative ultrastructural 3D analysis of 3 brain regions.

During my revision, I raised one major comment and several medium/minor comments that have been satisfactorily addressed by the authors.

In addition to my comments, two other reviewers (reviewers #1 and #2) raised concerns referring to different sections of the manuscript. In my opinion, the authors have addressed most of the comments indicated.

Overall, the revision performed has increased the quality of the manuscript, providing details of a new imaging and analysis tool that will provide extremely valuable information for the community.

My conclusion is that I recommend this paper for publication in Communications Biology.

Reviewers' comments:

Reviewer #1 (Remarks to the Author):

I acknowledge that the authors consider their work as validation of their workflow, rather than an analysis of the differences of the connectivity and synapses of different brain regions. However, this is not clearly expressed in the manuscript, except a sentence in the discussion. I agree that the validation of the workflow can be interesting per se, but the focus of the paper has to be changed, in my opinion.

1) If this is the aim of the manuscript, I think it should be clearly stated, starting from the abstract. In the current version of the paper, all the emphasis is still on the measurements, and the differences between brain regions. As the authors concurred, the volumes analyzed, are not significant for such a quantitative analysis.

2) As a proof-of-concept for a workflow, the authors should have an estimation of the accuracy of their approach, or at least discuss this point. I would also discuss more strengths and weaknesses of the approach.

On the other hand, I have appreciated the analysis of the non-synaptic fibers, which has improved the study, as well as the answers to all the other minor points i have raised in my previous report.

Reponse:

Regarding the aim of our manuscript and the role of the measurements that we provide, we wrote in the original version of the manuscript (Introduction, lines 147-154) the following paragraph:

“To explore the utility of this approach, we have formulated the following hypothesis: qualitative diversity in the cellular composition and connectivity of different brain regions should translate into quantitative differences in synaptic numbers and/or their distribution between dendrites and axons. To test this hypothesis, we have selected three regions of the adult mouse brain whose cellular composition and connectivity are very different: the stratum radiatum (SR) of the hippocampus (CA1 field), and layers 1 and 3 (L1 and L3) of the primary somatosensory cortex, in the hindlimb representation area (S1HL).”

We therefore think that the data we provide, and the analyses and comparisons we have made, are fully justified to demonstrate that our method is feasible and reliable.

We thank the reviewer for pointing out that the method and the quantitative data it generates might not be clear in the abstract. We have therefore modified the abstract to better describe these points. To further clarify the aim of the study, we have explicitly stated that we investigated the three regions “to test this methodology”. Additionally, to comply with the reviewer’s comment, we have changed the sentence:

“The quantitative data obtained through this method reveal a distinctive synaptic organization in each region, and enables us to identify subtle traits and differences that might have been overlooked in a qualitative analysis”

to:

“The quantitative data obtained through this method enable us to identify subtle traits and differences in the synaptic organization of the samples, which might have been overlooked in a qualitative analysis.”

In summary, the old abstract:

“We have traced all nerve fibers present within different samples of brain tissue using volume electron microscopy and dedicated software. Every fiber “skeleton” is linked to its corresponding synaptic contacts, so the result is an intricate meshwork of axons and dendrites interconnected by a cloud of synaptic junctions. With this approach, we have investigated quantitative differences between the stratum radiatum of the hippocampus and layers 1 and 3 of the somatosensory cortex. We have found that nerve fibers are densely packed in the neuropil, reaching up to 9 kilometers per cubic mm. From each of these regions we have obtained the number of synapses, the number and lengths of axons and dendrites, and the distributions of excitatory and inhibitory synapses on dendritic spines and shafts. The quantitative data obtained through this method reveal a distinctive synaptic organization in each region, and enables us to identify subtle traits and differences that might have been overlooked in a qualitative analysis.”

has been rewritten, and now reads:

“We have traced all nerve fibers present within different samples of brain tissue using volume electron microscopy and a specifically developed software tool. With this tool, individual dendrites and axons are traced, obtaining a simplified “skeleton” of each fiber, which is linked to its corresponding synaptic contacts. The result is an intricate meshwork of axons and dendrites interconnected by a cloud of synaptic junctions. To test this methodology, we have applied it to the stratum radiatum of the hippocampus and layers 1 and 3 of the somatosensory cortex. We have found that nerve fibers are densely packed in the neuropil, reaching up to 9 kilometers per cubic mm. We have obtained the number of synapses, the number and lengths of dendrites and axons, the linear densities of synapses established by dendrites and axons, and their location on dendritic spines and shafts. The quantitative data obtained through this method enable us to identify subtle traits and differences in the synaptic organization of the samples, which might have been overlooked in a qualitative analysis”

Regarding the accuracy of our approach, we have given a measure of the spatial resolution of the stacks of images (5 x 5 x 20 nm) in the Materials and Methods section (lines 608-609). In the Discussion (lines 433-441), we discuss the importance of using this relatively high resolution, since it allows us to identify and classify excitatory and inhibitory synapses based on precise ultrastructural features.

With respect to the strengths and weaknesses of our approach, a similar suggestion has also been made by reviewer #2. We had already discussed other methods in lines 405-412 and 431-434. To further clarify this point, we have modified and extended the two last paragraphs of the Discussion.

The original paragraphs, beginning in line 541:

“However, our present results should be considered with caution, since they represent a single snapshot of three different regions and do not account for the local or interindividual variability described in previous studies^{48,54,55}. A comprehensive study of any region, or a comparison between different regions, would require the analysis of multiple volumes. Our current methodological approach facilitates this process by offering detailed quantitative information in a robust and reliable way.

Finally, the strategy we propose for quantitatively examining synaptic connectivity in different brain regions may prove particularly useful and reliable compared to other connectomics studies. This approach can offer a better understanding of the structure and dynamics of the brain, as well as the opportunity to explore how they change during both normal and pathological conditions.”

have been modified and extended, and now read:

“However, our present results should be considered with caution, since they represent a single snapshot of three different regions and do not account for the local or interindividual variability described in previous studies^{48,54,55}. A comprehensive study of any region, or a comparison between different regions, would require the analysis of multiple volumes of brain tissue. Our current methodological approach facilitates this process by offering detailed quantitative information in a robust and reliable way. This approach can offer a better understanding of the structure and dynamics of the brain, as well as the opportunity to explore how these aspects change during both normal and pathological conditions.

Finally, the strategy we propose for quantitatively examining synaptic connectivity in different brain regions has the potential to be particularly useful and reliable compared to other connectomics studies. The key strength of our method is its simplicity, since we leverage a schematic representation of neuronal processes and their connections. This approach significantly reduces the time and computational resources required compared to dense reconstructions, while still providing valuable quantitative data about the connectivity within a specific region. A further benefit of our approach is its compatibility with other methods. This implies that information about the caliber of fibers or the volume fractions occupied by different tissue components, which are not directly available from the skeletons, can still be obtained through other means, including dense reconstructions.”

We have also updated reference 46 and the Acknowledgements section.

Reviewer #2 (Remarks to the Author):

I am satisfied by the new version of the manuscript. The authors carefully considered the questions that I asked and provided convincing answers.

I would have an additional comment and recommendation to make now that I have read through the paper once again:

One important message conveyed in the introduction is that this work provides a new and more efficient way to measure the connectome, diverging from the single-animal, exhaustive way of working that is taken by many in the field. I am missing a paragraph in the discussion that would come back to this. Maybe even the last paragraph. How does this new study address the difficulties of dense reconstructions? Where is the gain of the new technique? Is it faster? The speed of analysis, maybe also its computational load should be put in perspective of other methods.

Response:

We think that the main advantage of our method is its simplicity, since any fiber can be traced, measured and connected without the need to fully reconstruct it. This saves time and computer resources, but does not imply that our method opposes —or is incompatible with— dense reconstructions or any other method. We have tried to clarify this by modifying and extending the last two paragraphs of the Discussion. A similar question was raised by reviewer #1, so please see our response above.

We have also updated reference 46 and the Acknowledgements section.

Reviewer #3 (Remarks to the Author):

In the paper Turegano- Lopez et al, the authors provided results from a descriptive quantitative ultrastructural 3D analysis of 3 brain regions.

During my revision, I raised one major comment and several medium/minor comments that have been satisfactorily addressed by the authors.

In addition to my comments, two other reviewers (reviewers #1 and #2) raised concerns referring to different sections of the manuscript. In my opinion, the authors have addressed most of the comments indicated.

Overall, the revision performed has increased the quality of the manuscript, providing details of a new imaging and analysis tool that will provide extremely valuable information for the community.

My conclusion is that I recommend this paper for publication in Communications Biology.

Response:

We thank the reviewer for her/his comments. We think that they have helped us to improve the manuscript. Please also see the last modifications that we have made to the Abstract and Discussion, in response to the other two reviewers.